# Splitted Wavelet Differential Inclusion for Neural Signal Processing

## Abstract

Wavelet shrinkage is a powerful tool in neural signal processing. It has been applied to various types of neural signals, such as non-invasive signals and extracellular recordings. For example, in Parkinson's disease (PD), $\beta$ burst activities in local field potential signals indicate pathological information, which corresponds to *strong signal* with higher wavelet coefficients. However, it has been found that there also exists *weak signal* that should not be ignored. This weak signal refers to the set of small coefficients, which corresponds to the non-burst/tonic activity in PD. While it lacks the interpretability of the strong signal, neglecting it may result in the omission of movement-related information during signal reconstruction. However, most existing methods are mainly focused on strong signals while ignoring weak signals. In this paper, we propose *Splitted Wavelet Differential Inclusion*, which is provable to achieve better estimation of both the strong signal and the whole signal. Equipped with an $\ell_2$ splitting mechanism, we derive the solution path of a couple of parameters in a newly proposed differential inclusion, of which the sparse one can remove bias in estimating the strong signal and the dense parameter can additionally capture the weak signal with the $\ell_2$ shrinkage. The utility of our method is demonstrated by the improved accuracy in a numerical experiment and additional findings of tonic activity in PD.

## 1 Introduction

Neural signals are fluctuations in neuronal activity that play a crucial role in brain function and communication Buzsáki et al. (2012). To effectively preprocess these signals, wavelet analysis has emerged as a powerful tool and has been widely applied across various neuroscience analysesPavlov et al. (2012), as wavelet functions proficiently characterize transient fluctuations in rhythmic neural oscillations Luo et al. (2018); Donoghue et al. (2020). A typical example is to extract these neural oscillations at different frequency bands (e.g., $\alpha$ (8-12 Hz), $\beta$ (12-35 Hz), and $\gamma$ (35-100 Hz) bands) that are particularly important in neuropsychiatric disorders Mathalon & Sohal (2015) and movement disorders Yin et al. (2021). For such tasks, wavelet analysis facilitates the detection and quantification of temporal dynamics among neuronal populations, as observed in epilepsy Kalbhor & Harpale (2016) and Parkinson's Disease (PD) Lofredi et al. (2023); Tinkhauser et al. (2017b).

Of particular paramount application of wavelet analysis is Wavelet Shrinkage Donoho & Johnstone (1994; 1995; 1998); Donoho (1995), which is often employed to neural signal denoising Baldazzi et al. (2020). It projects the noisy data into the wavelet domain, followed by a hard or soft thresholding method to force noisy coefficients to zeros. Apart from denoising, Wavelet Shrinkage is also employed in decoding brain states from neural signals, particularly in subthalamic nucleus (STN) local field potential (LFP) signals in PD Khawaldeh et al. (2020); Tinkhauser et al. (2017b). These findings also support improved clinical efficacy in PD by an adaptive deep brain stimulation algorithm involving thresholds in Wavelet Shrinkage Nie et al. (2021); Guo et al. (2024). To select the threshold parameter, many methods have been proposed, such as *Universal* ($\sigma\sqrt{2\log n}$) Donoho & Johnstone (1994), Minimaxi Verma & Verma (2012), *SureShrink* Donoho & Johnstone (1995), Bayesian Shrinkage Do & Vetterli (2002); Simoncelli & Adelson (1996); JOHNSTONE & SILVERMAN (2005), non-parametric shrinkage Antoniadis & Fan (2001); Gao (1998) such as SureLet Luisier et al. (2007) and Neigh Shrink Sure Chen et al. (2005). Most of these methods only selected large coefficients since they are assumed to be all the information contains in the signal Atto et al.

(2011). However, it has been found that there may exist small coefficients Donoho & Johnstone (1994), such as the non-burst activity in PD Khawaldeh et al. (2020).

In this paper, we call the signal composed by such small (*resp.* large) coefficients as the weak (*resp.* strong) signal. Although the strong signal typically has more semantic meaning and is more interpretable than the weak signal, ignorance of the latter can lead to the loss of information in reconstruction. For example, exaggerated $\beta$ oscillations (10-35 Hz), especially low $\beta$ oscillations (10-20 Hz) in STN are significant biomarkers in the Parkinson's study Kühn et al. (2006); Lofredi et al. (2023), but the non-burst activity restricts the capacity of encoding physiological information in the basal ganglia network Khawaldeh et al. (2020); Brittain & Brown (2014); Mallet et al. (2008). At the mesoscopic scale, strong beta activities essentially come from the prolonged synchronized state between neurons Tinkhauser et al. (2017a) and these non-burst activities coming from low-frequency synchronized states might contain more information on motor movement Cole et al. (2017); Khawaldeh et al. (2020). Therefore, it is desirable to identify the strong signal as a semantic part and meanwhile estimate the whole signal (including both strong and weak signals) well.

Towards this goal, existing methods with too small or too large thresholds either suffer from failing to eliminate noise components in estimating the strong signal or ignoring the weak signal part in estimating the whole signal. Specifically, the methods with too large threshold values (*e.g.*, Universal, or Minimaxi) could eliminate the noise components, but they could induce large bias/errors in estimating the strong signal; moreover, they might over-smooth the weak signal part with small coefficients. On the other, the methods with smaller thresholds (*e.g.* SureShrink Donoho & Johnstone (1995)) might fail to eliminate noise components in estimating the strong signal for interpretation.

To resolve these problems, we propose a new method from the perspective of differential inclusion, dubbed as *Splitted Wavelet Differential Inclusion*(SWDI), which is provable to achieve better estimation than Wavelet Shrinkage on both the strong signal and the whole signal. Specifically, we introduce an $\ell_2$ splitting mechanism appended to the signal reconstruction loss, which couples a sparse parameter and a dense parameter to estimate the strong signal and the whole signal, respectively. To update these parameters, we propose a differential inclusion with sparse regularization, *i.e.*, a dynamic approach with a unique closed-form solution. Equipped with an early stopping mechanism on this dynamics, the sparse parameter can recover the strong signal without bias, while the dense parameter can additionally capture the weak signal via $\ell_2$ shrinkage. We also provide a discretization method that can efficiently generate the solution path, even if the wavelet decomposition matrix is not orthogonal. The utility and effectiveness of our method are demonstrated in a numerical experiment and the effectiveness study of dopaminergic medication impact on PD. Particularly, the signal recovered by our method is more significantly correlated to the medication, which can be explained by the non-burst activity found by our method.

To summarize, our contributions are listed as follows:

- We propose the *SWDI*, which involves a dual parameter to simultaneously estimate the strong and the whole signal.

- We theoretically show that the closed-form solution path can achieve better estimation than Wavelet Shrinkage on both the strong signal and the whole signal. For the strong signal, our estimation is bias-free, while for the whole signal, our dense parameter can accurately capture the weak signal via an $\ell_2$ shrinkage.

- We apply our method to neural signal recognition in PD and identify the non-burst activity that has been recently found to be also responsive to the medication.

## 2 RELATED WORK

**Wavelet Shrinkage.** The threshold selection has been a challenging problem for Wavelet Shrinkage methods, which can be traced back to Donoho & Johnstone (1994) that proposed a universal selection parameter $\sigma\sqrt{2\log n}$ where $n$ denotes the length of the signal and $\sigma$ denotes the noise level. Although it can eliminate noise components, it can induce biases/errors in estimation. To address this issue, many non-adaptive and adaptive methods have been proposed, such as Minimaxi Verma & Verma (2012), *SureShrinkage* Donoho & Johnstone (1995) that leveraged the Jame-Stein Shrinkage method for more accurate estimation, Bayesian shrinkage Do & Vetterli (2002); Simoncelli & Adelson (1996); JOHNSTONE & SILVERMAN (2005); ter Braak (2006), non-parametric shrink-

age Antoniadis & Fan (2001); Gao (1998) including SureLet Luisier et al. (2007) and Neigh Shrink Sure Chen et al. (2005), *etc.*

**Wavelet analysis in neural signal processing.** Wavelet analysis has been widely applied in processing brain signals Li et al. (2007); Faust et al. (2015); Ortiz-Rosario et al. (2015). One notable example is its use in establishing biomarkers for PD Wang et al. (2004); Tinkhauser et al. (2018); Luo et al. (2018); Nie et al. (2021); Guo et al. (2024). Among these biomarkers, substantial $\beta$ burst activity have been clinically demonstrated to correlate with improvements in mobilityTinkhauser et al. (2017b); Lofredi et al. (2023).

Most existing Wavelet Shrinkage methods relied on the *strongly sparsity* assumption Atto et al. (2011), *i.e.*, the signal (or each sub-band) is a representation of only a small proportion of strong signal coefficients with large magnitude. However, as we will show below, there may exist small/weak coefficients in many applications including neural signal processing, and should not be ignoredKhawaldeh et al. (2020).

**Weak signal coefficients** refer to those small coefficients that have been found in many applications, such as textures, contours in image denoising Atto et al. (2011), enlarged gray matter voxels in brain diseases Sun et al. (2017), the non-burst component in dopamine-dependent motor symptoms with Parkinson's patients Khawaldeh et al. (2020); Brittain & Brown (2014); Mallet et al. (2008). In these applications, these weak signal coefficients may not be as interpretable as strong signal coefficients (*e.g.*, burst component in PD analysis); however, the ignorance of these weak signals due to oversmoothing may lose information in signal reconstruction in neuroscience analysis. Therefore, it is desired to **i)** disentangle the strong signals as the semantic component of the signal, and meanwhile **ii)** accurately estimate the whole signal by capturing the weak signal coefficients.

**Limitations of Wavelet Shrinkage and our specifications.** Existing Wavelet Shrinkage methods either suffered from failing to disentangle strong signals apart because of small thresholds (*e.g.*, SureShrink), or ignoring weak signals because of large thresholds (*e.g.*, Minimaxi or Universal $\sqrt{2 \log n}$). **In contrast**, the splitting mechanism in our *Wavelet Differential Inclusion*(WDI) decouples sparse and dense parameters, enabling accurate estimation of both the strong signal and weak signal coefficients. Note that our method is motivated by but different from the differential inclusion method in signal recovery Osher et al. (2005; 2016). For the reason of coherence and space limit, we leave the review of these methods in Appx. A.

## 3 PRELIMINARY

**Problem setup.** Suppose we observe data $\{y_i\}_{i=1}^n$ with $n = 2^J$ for some integer $J > 1$, such that $y_i = f(t_i) + e_i$ with $e_1, ..., e_n \sim_{i.i.d} \mathcal{N}(0, \sigma^2)$, $t_i := \frac{i}{n}$ and $f$ denoting the ground-truth signal we would like to recover. One may construct a wavelet transformation to decompose the observed signal $y$ into an orthogonal wavelet basis, including *stationary wavelet transform* (SWT), *discrete wavelet transform* (DWT), *etc.* For DWT, we obtain an inverse wavelet transform matrix $W \in \mathbb{R}^{n \times n}$ depending on the type of wavelet filters (such as Coiflets, Symlets, Daubechies Cohen et al. (1993), Beylkin Beylkin et al. (1991), Morris minimum-bandwidth Morris & Peravali (1999)), the number of vanishing moments $M$ and the coarsest resolution level $L$.

With such $W$, we can obtain the coefficients $\theta$ up to the linear transformation of noise:

$$\omega = \theta^* + \varepsilon, \ \omega = Wy, \ \varepsilon = We \sim \mathcal{N}(0, \sigma^2 I_n),$$

when $W$ is orthogonal. Here, we assume the ground-truth signal $\theta^*$ contains three types of elements:

1. *Strong signal set.* $S := \left\{ i : |\theta_i^*| > 2\sigma(1 + a)\sqrt{2 \log n} \right\}$ for some constant $a > 0$.
2. *Weak signal set.* $T := \{ i : 1 < |\theta_i^*| = o(\sqrt{2 \log n}) \}$.
3. *Null set.* $N := \{ i : \theta_i^* = 0 \}$.

The strong signal set corresponds to elements with large magnitudes and can be identified by the Wavelet Shrinkage Donoho & Johnstone (1995). This definition aligns with the $\beta$-*min* condition Zhang & Zhang (2014) in variable selection, meaning that the signal should be strong[1] enough to be identified. Correspondingly, we define the strong signal vector $\theta^{*,s}$ as $\theta_i^{*,s} = \theta_i^*$ for $i \in S$;

---

[1]In linear regression, it requires $\theta_i^* \succeq O(\frac{\log n}{p})$ for $\theta_i^* \neq 0$.

= 0 otherwise. The weak signal set refers to those non-zero elements with smaller magnitudes and has been similarly introduced in wavelet denosing Atto et al. (2011) and beyond Li et al. (2019); Zhao et al. (2018). Despite with small magnitudes, this set may have a non-ignorable impact on the reconstruction effect. However, it can be difficult for existing $\ell_1$-regularization (*e.g.*, the Wavelet Shrinkage) to distinguish them from the Null set.

**Wavelet shrinkage via soft-thresholding.** The Wavelet Shrinkage method in Donoho & Johnstone (1994); Donoho (1993); Donoho & Johnstone (1995) proposed the soft-thresholding estimator $\widehat{\theta}_i = \eta(\omega_i, \lambda) = \text{sign}(\omega_i) \max(|\omega_i| - \lambda, 0)$ for some $\lambda > 0$, followed by inverse wavelet transformation to recover the signal $\widehat{f} := W^{-1}\eta(\omega, \lambda)$. To remove the noise components, Donoho & Johnstone (1994) selected $\lambda \sim O(\sqrt{2 \log n})$, which is provable to be minimax optimal.

**Proposition 3.1** (Theorem 2 in Donoho & Johnstone (1994))**.** *Denote $\widehat{\theta}(\lambda_n) := \eta(\omega, \lambda_n)$, then the minimax threshold $\lambda_n^* := \arg\inf_{\lambda \geq 0} \sup_{\theta^*} \mathbb{E}[\|\widehat{\theta}(\lambda) - \theta^*\|_2^2] \sim \sqrt{2 \log n}$.*

Although $\lambda \sim \sqrt{2 \log n}$ can effectively remove noise with high probability, it suffers from two limitations: **i)** the estimation of strong signal coefficients is biased due to non-zero $\lambda$; **ii)** it ignores the weak signal coefficients, which leads to additional errors in estimating the whole signal $\theta^*$. Specifically, for **i)**, although the threshold $\sqrt{2 \log n}$ can identify the strong set $S$ by removing others with high probability, it can induce bias in estimating $\theta^{*,s}$. This is shown by the following result, which states that once we identify $S$, the optimal threshold value in estimating $\theta^{*,s}$ is $\lambda = 0$.

**Proposition 3.2.** *For strong and weak signals,* i.e., $i \in S \cup T$ with $|\theta_i^*| > 1$, *we have:*

$$0 = \arg\inf_{\lambda} \sup_{|\theta_i^*| \geq 1} \mathbb{E}(\eta(\omega_i, \lambda) - \theta_i^*)^2.$$

*Remark* 3.3. This result shows that in terms of population error, the best optimal threshold value is also 0 for weak coefficients. However, it does not mean we should select $\lambda = 0$ to estimate $\theta^*$. First, it fails to remove noise components in $N$. Moreover, even for weak signals, we will show that for any fixed $\theta^*$, applying an appropriate non-zero $\ell_2$ shrinkage would achieve better estimation.

For **ii)**, $\lambda \sim \sqrt{2 \log n}$ fails to account for weak signal coefficients that are $o(\sqrt{2 \log n})$. To achieve a more accurate estimation, Donoho & Johnstone (1995) proposed *SureShrink* (Stein's unbiased estimate of risk), which can reduce biases in $\theta^*$. However, it can mistakenly induce noise and weak signals in identifying the strong signal $\theta^{*,s}$. This may be undesired in neuroscience analysis where identifying the strong signal is important for interpretation, such as the bust component identification in Biomarker identification in Parkinson's patients.

In summary, previous methods either have a bias of the strong signal and ignore the weak in estimating $\theta^*$; or fail to remove non-signal coefficients in estimating $\theta^{*,s}$.

## 4 SPLITTED WAVELET DIFFERENTIAL INCLUSION

We introduce a new method from the perspective of *differential inclusion*[2], which can simultaneously identify the strong signal $\theta^{*,s}$ and accurately estimate the whole signal $\theta^*$. In Sec. 4.1, we first introduce the WDI that can remove bias in estimating $\theta^{*,s}$, followed by Sec. 4.2 where an $\ell_2$ splitting mechanism is additionally introduced to capture both $\theta^{*,s}$ and $\theta^*$.

### 4.1 WAVELET DIFFERENTIAL INCLUSION FOR $\theta^{*,s}$

To estimate the strong signal $\theta^{*,s}$, we introduce the squared error loss $\ell(\theta) := \frac{1}{2}\|\omega - \theta\|_2^2$ and consider the following differential inclusion with an $\ell_1$ regularization:

$$\dot{\rho}(t) = -\nabla_\theta \ell(\widehat{\theta}(t)) = \omega - \theta(t), \tag{1a}$$

$$\rho(t) \in \partial\|\theta(t)\|_1, \tag{1b}$$

where $\rho$ is the sub-gradient[3] of $\|\theta\|_1$ and $\rho(0) = \theta(0) = 0$. We call Eq. 1 as WDI. Note that it is a special form of the *Bregman Inverse Scale Space* (ISS) Osher et al. (2016) when the design matrix

---

[2]The ordinary differential inclusion is similar to the ordinary differential equation. For details about its role in signal recovery, please refer to related works in Appx. A.

[3]The sub-gradient of $f$ at $x_0$ is defined as $\partial f(x_0) := \{g : f(x) - f(x_0) \geq \langle g, f(x_0) \rangle$ for any x$\}$.

of $\theta$ is set to the identity matrix in the linear model, and can be viewed as a continuous dynamics of *Bregman Iteration* Osher et al. (2005) in image denoising. Thanks to the identity matrix design, the differential inclusion of Eq. 1 in the wavelet scenario has a closed-form solution $\theta(t)$, which can illustrate the effectiveness of bias removal in estimating $\theta^{*,s}$.

Specifically, starting from $\rho(0) = 0$, Eq. 1 generates a unique solution path of $\theta(t)$, in which more elements become non-zeros as $t$ grows, as shown below:

**Proposition 4.1.** *The solution of Eq. 1 is $\theta_j(t) = \omega_j(t)$ for $t \geq \frac{1}{|\omega_j|}$; and $= 0$ otherwise for each $j$. Therefore, $\theta_j(t)$ and $\rho_j(t)$ are right continuous w.r.t. $t$ for each $j$.*

*Remark* 4.2. To better explain the effect of bias removal, we compare Eq. 1 with Wavelet Shrinkage, whose solution $\theta(t)$ ($t = \frac{1}{\lambda}$) is equivalent to the Lasso estimator: $\frac{1}{2}\|\omega - \theta\|_2^2 + \frac{1}{t}\|\theta\|_1$. By taking gradient w.r.t. $\theta_t$, the solution satisfies $\frac{\rho(t)}{t} = \omega - \theta(t)$ with $\rho(0) = \theta(0) = 0$. When $|\rho_j(t)|$ becomes non-zero, then we have $\theta_j(t) = \omega_j - \frac{\rho_j(t)}{t}$, where $\frac{\rho_j(t)}{t}$ can induce the bias. **As a contrast**, when $|\rho_j(t)| = 1$ of WDI at some $t$, since $\theta_j(t)$ is right continuous w.r.t. $t$, we have that if $t'$ is in a small neighborhood of $t$, $\theta_j(t')$ is non-zero and has the same sign of $\theta_j(t)$. Therefore, we have $\dot{\rho}_j(t) = \lim_{t' \to t} \frac{\theta_j(t') - \theta_j(t)}{t' - t} = 0$, which gives $\theta_j(t) = \omega_j$ according to Eq. 1a.

As shown in Prop. 4.1, $t$ plays a similar role as $1/\lambda$ in disentangling the strong signal from others. However, it is interesting to note that, unlike the Wavelet Shrinkage, the solution $\theta_j(t) = \omega_j = \eta(\omega_j, \lambda = 0)$ is without additional threshold parameter! In contrast to $\theta_j(\lambda = \sqrt{2\log n}) := \eta(\omega_j, \sqrt{2\log n})$, this estimator can remove the bias caused by $\lambda$. Therefore, our differential inclusion can not only remove noise and weak components when $t$ is large enough but also can estimate $\theta^{*,s}$ without bias induced by $\lambda$ in Wavelet Shrinkage that is necessary for removing bias. Equipped with such a bias removal of Eq. 1, we can achieve a smaller $\ell_2$ error than the Wavelet Shrinkage.

**Theorem 4.3.** *Denote $\theta_{\min}^{*,s} := \min_{i \in S} |\theta_i^*|$ and $s := |S|$. Then at $\bar{\tau} := 1/((1 + a)\sqrt{2\log n})$ and $n$ is large enough such that $\frac{a}{2}\sqrt{2\log n} > \theta_j^*$ for $j \notin S$. Denote $\theta(t)$ as the solution of Eq. 1, then with probability at least $1 - 2n^{-4a^2} - \max\left(\exp\left(-s\lambda^2/8\right), n^{-(1+a)^2/4}\right)$, we have*

$$\|\theta(\bar{\tau}) - \theta^{*,s}\|_2 < \|\eta(\omega, \lambda) - \theta^{*,s}\|_2 \ \text{for any } \lambda > 0. \tag{2}$$

*Remark* 4.4. The proof of Thm. 4.3 is left to Appx. D. We will show that stopping at $\bar{\tau}$ will remove other components since $\max_j |\varepsilon_j| < (1 + a)\sqrt{2\log n}$ with high probability.

Recall that $\eta(\omega, \lambda)$ refers to the soft-thresholding solution of Wavelet Shrinkage. Thm. 4.3 means that the WDI gives a more accurate estimation than Wavelet Shrinkage.

### 4.2 WAVELET DIFFERENTIAL INCLUSION WITH $\ell_2$-SPLITTING FOR BOTH $\theta^{*,s}$ AND $\theta^*$

In this section, we proceed to capture the weak signal, in order to estimate the whole signal $\theta^*$ more accurately. To achieve this, we propose the *SWDI*, which generates a solution path of a sparse parameter $\theta^s(t) \in \mathbb{R}^n$ coupled with a dense parameter $\theta(t)$ introduced by an $\ell_2$ splitting term. We will show that the $\theta^s(t)$ maintains the same bias removal property as WDI in Eq. 1; moreover, the dense parameter can additionally capture the weak signal with the $\ell_2$ shrinkage induced by the $\ell_2$ splitting mechanism. Specifically, we consider the loss $\bar{\ell}_\nu(\theta, \theta^s) := \frac{1}{2}\|\omega - \theta\|_2^2 + \frac{\nu}{2}\|\theta - \theta^s\|_2^2$, where $\frac{\nu}{2}\|\theta - \theta^s\|_2^2$ with $\nu > 0$ denotes the $\ell_2$ splitting term that introduces a couple of parameters, which is expected to simultaneously estimate the strong signal $\theta^{*,s}$ and the whole signal $\theta^*$ well. This is achieved by the following differential inclusion:

$$0 = -\nabla_\theta \bar{\ell}_\nu(\theta, \theta^s) = \omega - (1 + \nu)\theta(t) + \nu\theta^s(t), \tag{3a}$$

$$\dot{\rho}(t) = -\nabla_{\theta^s} \bar{\ell}_\nu(\theta, \theta^s) = \nu(\theta(t) - \theta^s(t)), \tag{3b}$$

$$\rho(t) \in \partial\|\theta^s(t)\|_1, \tag{3c}$$

where $\rho(0) = \theta(0) = \theta^s(0)$. Similar to the WDI in Eq. 1, when $t$ is large enough, it can remove the noise and weak signal components, *i.e.*, $T \cup N$ to identify the strong signal component in $\theta^s$; whereas the parameter $\theta$ can additionally capture weak components as it is dense. Formally, this property can be shown by the following solution path of Eq. 3:

**Proposition 4.5.** *The solution of differential inclusion in Eq. 3 is*

$$
\begin{cases}
\theta_j(t) = \theta_j^s(t) = \omega_j, & t \geq \frac{1+1/\nu}{\omega_j} \\
\theta_j(t) = \frac{\omega_j}{1+\nu}, \ \theta_j^s(t) = 0 & t < \frac{1+1/\nu}{\omega_j}
\end{cases} \forall j.
$$

Prop. 4.5 suggests that when $t$ is large enough, the noise and weak components can be removed in $\theta^s$ and meanwhile, which is the same as the solution in WDI in Eq. 1, can estimate the strong components without bias. On the other hand, for the dense parameter $\theta(t)$, it keeps the strong components in $\theta^s$ and meanwhile estimates the weak signals with $\frac{\omega_j}{1+\nu}$ via the $\ell_2$ shrinkage.

To explain in more detail, note that for the strong signal, when $|\rho_j(t)| = 1$, we similarly have $\dot{\rho}_j(t) = 0$ according to remark 4.2. Therefore, we have $\theta_j^s(t) = \theta_j(t) = \omega_j$ according to Eq. 3b. When $|\rho_j(t)| < 1$ for some $j$, we have $\theta_j^s = 0$ and Eq. 3a gives $\theta_j = \frac{\omega_j}{1+\nu}$. For the whole signal $\theta$, note that $\theta_j(t)$ at $t > t_j$ given by Eq. 3a is the minimizer of $\frac{1}{2}(\omega_j(t) - \theta_j(t))^2 + \frac{\nu}{2}(\theta_j(t))^2$, where $\frac{\nu}{2}(\theta_j)^2$ can be viewed as an $\ell_2$ regularization of $\theta_j$. This $\ell_2$ shrinkage, which is equivalent to the *maximum a posteriori* (MAP) estimate with the Gaussian prior $\nu \sim \mathcal{N}(0,1)$, resembles the shrinkage effect in the Jame-Stein Estimator. With this shrinkage, we will show that $(\theta^s(t), \theta(t))$ can estimate the $\theta^{*,s}$ and $\theta^*$ well.

**Theorem 4.6** (Informal). *Denote $\theta_{\max,T}^* := \max_{i \in T} |\theta_i^*|$ and $n$ is large enough such that $\theta_{\max,T}^* < a_0\sqrt{\log n}$ for some $0 < a_0 < 1$. Then for $(\theta(t), \theta^s(t))$ in Eq. 3, if $n > 4^{1/(1-a_0)}$ at $\bar{\tau} := (1 + \frac{1}{\nu})/((1+a)\sqrt{2\log n})$, the following holds with probability at least*

$$
1 - O\left(n^{-4a^2}\right) - O\left(n^{-s/32}\right) - O\left(n^{-(1+a)^2/16}\right) - O\left(\exp(-|T|)\right) - O\left(\exp(-|N|)\right),
$$

1. **Strong signal recovery.** *For the strong signal $\theta_S^{*,s}$,*

$$
\|\theta^s(\bar{\tau}) - \theta^{*,S}\|_2 < \|\eta(\omega, \lambda) - \theta^{*,s}\|_2, \ \forall \lambda > 0. \tag{4}
$$

2. **Weak signal recovery.** *For the weak signal $\theta_T^*$, there exists $\infty > \nu^* > 0$ such that*

$$
\|\theta(\bar{\tau})_T - \theta_T^*\|_2 < \|0 - \theta_T^*\|_2 = \|\theta_T^*\|_2 \ i.e., \ \nu = \infty \ shrinkage \ to \ 0;
$$
$$
\|\theta(\bar{\tau})_T - \theta_T^*\|_2 < \|\omega_T - \theta_T^*\|_2 = \|\varepsilon_T\|_2 \ i.e., \ \nu = 0 \ no \ shrinkage.
$$

3. **Whole signal recovery.** *For $\theta^*$, under the same $\nu^*$ in item 2, we have*

$$
\|\theta(\bar{\tau}) - \theta^*\|_2 < \|\eta(\omega, \lambda) - \theta^*\|_2, \ \forall \lambda \geq \sqrt{\log n}.
$$

Item 1 inherits the property in Thm. 4.3 for WDI in Eq. 1. Item 2 means we can better estimate the weak components $\theta_T^*$ via $\ell_2$ shrinkage. Finally, item 3 means our SWDI is more accurate than the Wavelet Shrinkage method with $\lambda \sim \sqrt{2\log n}$. Although this conclusion may not hold for $\lambda = o(\sqrt{\log n})$, the Wavelet Shrinkage with these $\lambda$'s fails to remove noise in identifying $\theta^{*,s}$.

**Selecting stopping time $\bar{\tau}$ and $\nu^*$.** The $\bar{\tau} := (1 + 1/\nu)/((1+a)\sqrt{2\log n})$ involves an unknown parameter $a$, which is used to define the level of strong signal coefficients $\theta^{*,s}$ Donoho & Johnstone (1994). Empirically, we can set it to a small constant $1 \geq a \geq 0$ so as to remove other components and identify as many strong components as possible. For $\nu^*$, we will show in Appx. E that $\nu^* := (\|\varepsilon_{T \cup N}\|_2^2 + \sum_{j \in T} \theta_j^* \varepsilon_j)(\|\theta_T\|_2^2 + \sum_{j \in T} \theta_j^* \varepsilon_j) \sim n/(|T|\mathrm{mean}(\theta_T^*))$. This term is approximate $O(1)$ since the weak signal is typically very dense (*i.e.*, $\frac{|T|}{n} \sim O(1)$). For example, the non-burst activity in the low $\beta$ band should be dense enough to be responsive to the medication.

**Solution path generation via linearization.** One can generate a solution path according to Prop. 4.5. Here we consider another method to generate $(\theta^s(t), \theta(t))_t$ via linearization proposed in Yin et al. (2008). Specifically, we consider the following differential inclusion:

$$
\frac{\dot{\theta}(t)}{\kappa} = -\nabla_\theta \bar{\ell}_\nu(\theta, \theta^s) = \omega - (1+\nu)\theta(t) + \nu\theta^s(t), \tag{5a}
$$

$$
\dot{v}(t) = -\nabla_{\theta^s} \bar{\ell}_\nu(\theta, \theta^s) = \nu(\theta(t) - \theta^s(t)), \tag{5b}
$$

$$
v(t) \in \partial\left(\|\theta^s(t)\|_1 + \frac{1}{2\kappa}\|\theta^s(t)\|_2^2\right) = \rho(t) + \frac{\theta^s(t)}{\kappa}, \tag{5c}
$$

where we introduce an $\ell_2$ norm $\frac{1}{2\kappa}\|\theta^s\|_2^2$ for discretization, with $\kappa > 0$ denoting the damping factor. We show in Appx. E that the solution $(\theta^{s,\kappa}(t), \theta^\kappa(t))$ of Eq. 5 converges to $(\theta^s(t), \theta(t))$ in Eq. 3 as $\kappa \to \infty$. Therefore, Thm. 4.6 still holds in Eq. 5 when $\kappa$ is large enough. By approximating $\frac{\dot{\theta}(t)}{\kappa}$ and $\dot{v}(t)$, we have the following discrete solution of Eq. 5:

$$\theta(k+1) = \theta(k) + \kappa\delta\left(\omega - (1+\nu)\theta(k) + \nu\theta^s(k)\right), \tag{6a}$$

$$v(k+1) = v(k) + \delta\nu(\theta(t) - \theta^s(t)), \tag{6b}$$

$$\theta^s(k+1) = \kappa\eta(v(k+1), 1), \tag{6c}$$

where $\delta$ denotes the step size. We show in Appx. F with sufficiently large $\kappa$, Thm. 4.6 still holds in Eq. 6 as long as $\delta < 2/(\kappa(1+\nu))$. Note that this discrete form can be extended to solve the differential inclusion for the general objective function $\bar{\ell}_\nu(\theta, \theta^s) := \frac{1}{2}\|y - W\theta\|_2^2 + \frac{\nu}{2}\|\theta - \theta^2\|_2^2$, where the wavelet matrix may not be orthogonal and we do not have a closed-form solution.

**Selecting damping factor $\kappa$.** $\kappa$ trade-offs between estimation accuracy and computational efficiency. In Appx. E, we show that as $\kappa \to \infty$, the solution $(\theta^{s,\kappa}(t), \theta^\kappa(t))$ of Eq. 5 converges to $(\theta^s(t), \theta(t))$. On the other hand, since $\delta$ is inversely proportional to $\kappa$, a larger value of $\kappa$ will take more iterations to find $\bar{\tau}$. Empirically, we find that $\kappa = 20$ works well.

**Selecting step size $\alpha$.** As shown in Appx. E, as long as $\delta < 2/(\kappa(1+\nu))$, we have Thm. 4.6 for Eq. 6. Empirically, we set it to $\delta = 1/(\kappa(1+\nu))$.

## 5 NUMERICAL EXPERIMENTS

In this section, we apply our method to synthetic data in estimating both the sparse signal $\theta^{*,s}$ and the whole signal $\theta^*$.

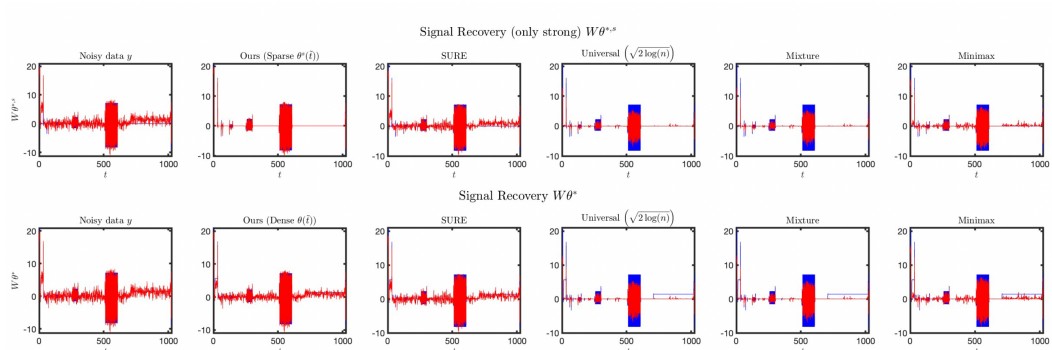

Figure 1: Visualization of Signal Recovery of $W\theta^{*,s}$ (top) and $W\theta^*$ (bottom). The blue curve represents the original signal ($W\theta^{*,s}$ in the top row and $W\theta^*$ in the bottom row), and the red curve represents the estimated one.

**Data generation.** We set $n = 1024$ and the 1-$d$ DWT matrix $W \in \mathbb{R}^{n \times n}$ as the Daubechies 6 with level 5, which is orthogonal. For the coefficients $\theta^* \in \mathbb{R}^n$, we respectively set the strong signal index set and the weak signal index set as $S := \{1, 4, 7, ..., 199\}$ and $T := \{401, 403, ..., n\}$. We set $a = 0.3$ and $\theta_i^* = 2(1+a)\sqrt{2\log(n)}$ if $i \in S$; $= 2$ if $i \in T$; and $= 0$ otherwise. The sequence $f = W\theta^* + \varepsilon$ with $\varepsilon_1, ..., \varepsilon_n \sim_{i.i.d} \mathcal{N}(0,1)$ is generated, and we report the $\ell_2$ error over 20 times in estimating $\theta^{*,s}$ and $\theta^*$.

**Implementation details.** We compare with the following threshold value methods that estimate $\hat{\theta} := \eta(W'f, \lambda)$, which includes **i) SURE** Donoho & Johnstone (1995) that selects $\lambda$ based on Stein's Unbiased Risk Estimate; **ii) Universal** method that constantly sets $\lambda = \sqrt{2\log(n)}$ Donoho & Johnstone (1994); **iii) Mixture** method that Verma & Verma (2012) combines **SURE** and **Universal**, depending on the signal-to-noise (SNR) ratio. Specifically, if the SNR is high, the **Universal** adopts the same threshold value with the **Universal** method; and **iv) Minimax** Verma & Verma (2012) that selects $\lambda$ using a minimax rule, *i.e.*, $\lambda = (0.3936 + 0.10829 \log_2 n)$ if $n > 32$ and $= 0$ otherwise.

For our method, we set $\kappa = 1000$, $\delta = \frac{1}{\kappa(1+\nu)}$, $\nu = \frac{1}{2}$ and the stopping time $\widehat{t} = \frac{1+1/\nu}{2\sqrt{2\log n}}$ s.t. our final estimations are $\theta(\widehat{t})$, $\theta^s(\widehat{t})$.

**Visualization of reconstructed signals.** As shown in Fig. 1, our method can well recover $W\theta^{*,s}$ (top row) and $W\theta^*$ (bottom row); as a contrast, SURE induces additional errors accounted by weak signals and noise in estimating $\theta^{*,s}$ while other methods with excessive shrinkage strategy will suffer from inaccurate estimations of $\theta^{*,s}$ and the ignorance of the weak signals in estimating $\theta^*$.

**Results analysis.** We report the relative $\ell_2$ error of $\theta^{*,s}$ and $\theta^*$ in Tab. 1. As shown, our method has a smaller error compare to others. Specifically, for the strong signal $\theta^{*,s}$, all methods except SURE adopt an overly large threshold value ($\sim O(\log n)$) which can induce errors in estimating strong signals. The SURE method with a smaller threshold value, however, induces noise components into the estimation, which can explain why SURE also suffers from a large error. For $\theta^*$, our method is comparable to the SURE and outperforms others that drop the weak signals.

Table 1: Average $\pm$ Std of relative $\ell_2$-error of $\theta^*$ and $\theta^{*,s}$.

| | Sparse Error ($\frac{\|\widehat{\theta}-\theta^{*,s}\|_2}{\|\theta^{*,s}\|_2}$) | Dense Error ($\frac{\|\widehat{\theta}-\theta^*\|_2}{\|\theta^*\|_2}$) |
|---|---|---|
| SURE | $0.4195 \pm 0.0180$ | $0.3001 \pm 0.0073$ |
| Universal ($\sqrt{2\log(n)}$) | $0.3991 \pm 0.0099$ | $0.5437 \pm 0.0060$ |
| Mixture | $0.3991 \pm 0.0099$ | $0.5437 \pm 0.0060$ |
| Minimax | $0.2849 \pm 0.0099$ | $0.4297 \pm 0.0058$ |
| **Ours ($\theta(\widehat{t})$)** | $0.4686 \pm 0.0114$ | $\mathbf{0.2918 \pm 0.0063}$ |
| **Ours ($\theta^s(\widehat{t})$)** | $\mathbf{0.1400 \pm 0.0373}$ | $0.3742 \pm 0.0064$ |

$\ell_2$ **error along the solution path.** As shown in Fig. 2, $\|\theta(t) - \theta^*\|_2$ (blue curve in the right) and $\|\theta^s(t) - \theta^{*,s}\|_2$ (red curve in the left) first decreases then increases as $t$ grows. For $\theta^{*,s}$, the $\theta^s(t)$ continuously identifies more signals until all strong signals are picked up. Meanwhile, the dense parameter $\theta(t)$ can additionally learn weak signals, therefore showing a smaller error in estimating $\theta^*$. If $t$ continues to increase, $\theta^s(t)$ will learn weak signals and finally both $\theta^s(t)$ and $\theta(t)$ converge to the noisy coefficients $W'y$. Moreover, our estimated stopping time $\widehat{t}$ (blue vertical line) yields a comparable estimation error to the minimum in the solution path (red vertical line).

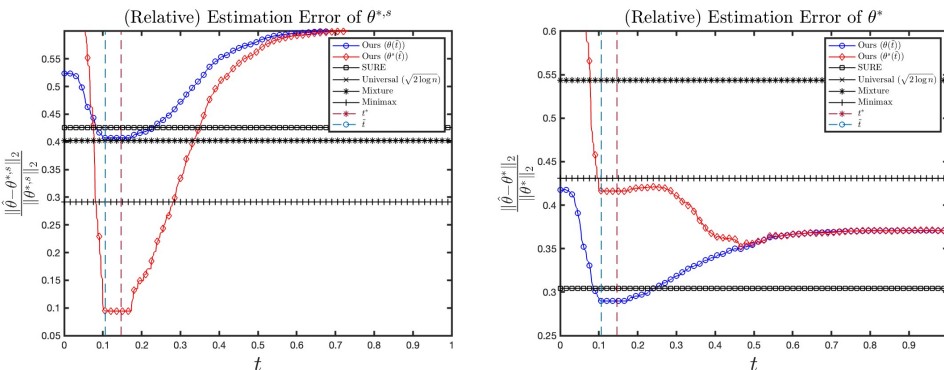

Figure 2: $\ell_2$ error of $\theta^{*,s}$ (left) and $\theta^*$ (right) along the path. The blue (*resp.* red) curve represents the MSE of $\theta(t)$ (*resp.* $\widetilde{\theta}(t)$). The blue (*resp.* red) vertical represents the estimated (*resp.* ground-truth) stopping time $\widehat{t}$ (*resp.*, $t^*$).

# 6 NEURAL SIGNAL RECOGNITION IN PARKINSONIANS

We apply our method to the signal neural signals reconstruction in PD[4], which distinguish pathological conditions in PD patients (ON/OFF medication).

---

[4]We also apply to Electroencephalography (EEG) signal denoising in Appx. H.

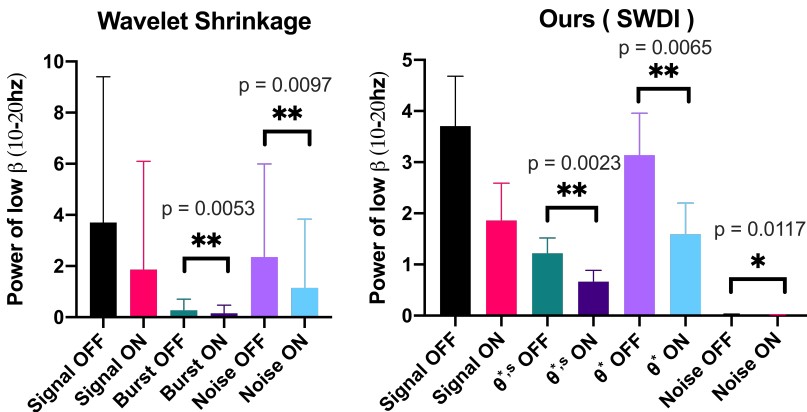

Figure 3: Two-sample T-test on the change of signal's energy after medication.

**Data & problem description.** We consider $\beta$ band LFP signals recorded in the STN Brown & Williams (2005), which are pathologically associated with the clinical condition denoted by the Unified Parkinson's Disease Rating Scale (UPDRS) in individuals with PD Maling & McIntyre (2016); Kühn et al. (2006); Little et al. (2013). For such signals, the $\theta^{*,s}$ corresponds to beta burst activity that contains prolonged synchronization activities from clusters of hundreds of neurons Tinkhauser et al. (2017b); Buzsáki et al. (2012) while $\theta^*$ contains both burst and tonic/non-burst activities. Although increased burst activity of $\beta$ band signals (especially low $\beta$ band Khawaldeh et al. (2022)) has been the most typical biomarker of PD Lofredi et al. (2023); Tinkhauser et al. (2017b), the non-burst signal may further help predict the upcoming movement Khawaldeh et al. (2020).

In total, 17 PD patients (32 hemispheres) are included in this study Nie et al. (2021); Wiest et al. (2023); Guo et al. (2024). Bipolar LFP signals of STNs are recorded in 17 patients (four females) with advanced PD who underwent the bilateral implantation of deep brain stimulation electrodes for clinical treatment before (*i.e.*, "OFF") and after (*i.e.*, "ON") taking levodopa. We adopt Guo et al. (2024) to process these signals. Artifacts, including large baseline shifts and muscle movement artefacts, are rejected through visual inspection. The recordings are further processed through a 90 Hz low-pass filter and a 2 Hz high-pass filter, followed by resampling at 320 Hz, therefore positioning low-beta oscillations into the fourth layer of stationary wavelet coefficients.

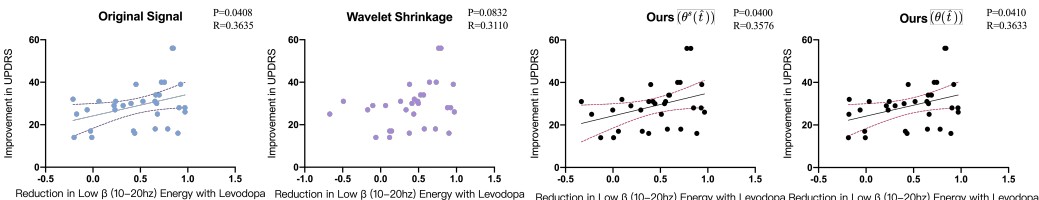

Figure 4: Correlation between changes of signal's energy and the UPDRS that measures the improvement of motor symptoms.

**Implementation details.** We follow Luo et al. (2018) to perform a 1-$d$ SWT on LFP signals as the Symlet 8 with level 6. We follow Donoho & Johnstone (1995) to estimate $\sigma$ as $\tilde{\sigma} = \text{Median}(W_j)/0.6745$. For Wavelet shrinkage, we select $\lambda$ according to the minimax rule in Donoho & Johnstone (1994). To well adapt to each layer, we follow Baldazzi et al. (2020) to multiply $\lambda$ with $1/(\ln j + 1)$ for the layer $j$. For SWDI, we set $\kappa = 20$, $\delta = 1/(\kappa(1 + \nu))$ with $\nu = 0.1$ and the stopping time as $\hat{t} = (1 + 1/\nu)/(\tilde{\sigma}/(\ln j + 1))$.

**Energy in ON vs. OFF medication.** With reconstructed signals by inverse wavelet transformation, we implement a two-sample T-test to measure whether the signal's energy is significantly reduced before and after receiving medications. Here, the energy is defined as the power of low $\beta$, *i.e.*,

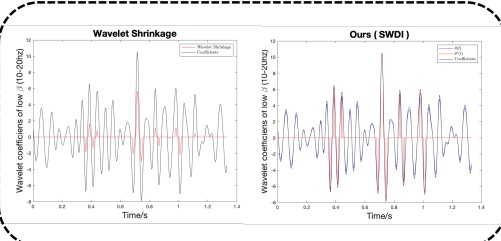 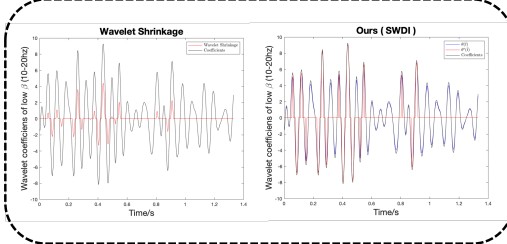

Figure 5: Estimated Signal Coefficients by Wavelet Shrinkage (left) and ours (right).

$E(t) = \frac{1}{\bar{w}} \sum_{i=t-\bar{w}+1}^{t} \widehat{f}^2(i)$ in Maling & McIntyre (2016), with $\bar{w}$ denoting the window size. We report the $p$-values in Fig. 3. As shown, the reconstructed strong signal (resp. correspond to "Burst" in Wavelet Shrinkage and "$\theta^{*,s}$" of our SWDI) of our method is more significant ($p = 0.0023$) than Wavelet Shrinkage ($p = 0.0053$) in the response to medications, which may due to the effectiveness in bias removal. On the other hand, the effect of the "noise" component is also significant ($p = 0.0117$), which can be attributed to the non-burst component that also exhibits a significant correlation, *i.e.*, $p = 0.0065$.

**Correlation with motor symptoms improvement.** To explain the energy reduction from signals recovered by SWDI, Fig. 4 reports the correlation between the "reduction in the low $\beta$ power" (*i.e.*, energy) and the improvement of motor symptoms measured by the change of clinical UPDRS score defined in Goetz et al. (2008). As shown, our reconstructed strong signal with $\theta^s$ is correlated to the improvement of motor symptoms ($p = 0.0400$, $R = 0.3576$) while the one given by Wavelet Shrinkage is not significant ($p = 0.0832$, $R = 0.3110$). Besides, with additional learned non-burst components which are recognized as noise by Wavelet Shrinkage, the $\theta(\hat{t})$ shows an even stronger correlation ($p = 0.0410$, $R = 0.3633$).

**Results analysis.** Fig. 3,. 4 suggest that the reconstructed signals by our method contain more physiological and clinical information. Specifically, the whole signal we learn is not only composed of synchronized/burst activity (in the Low $\beta$ band) that corresponds to high amplitude components; but also other lower components called the non-burst activity. Traditional methods mainly focus on the effectiveness of medication in inhibiting burst activity Lofredi et al. (2023); Tinkhauser et al. (2017b); while our method additionally shows that such an inhibition also happens on non-burst activity, which echos a recent finding in Khawaldeh et al. (2020). To explain how such a non-burst activity affects the LFP signals, a recent study Kajikawa & Schroeder (2011) hypothesized that such a non-burst activity may correspond to the electric field environment of neuron clusters. Since the LFP signal, which appears as a mixture of local potentials from neuron clusters, has been found to be affected by the fluctuation of this field environment Caruso et al. (2018), the change of this non-burst activity after medications may lead to the change of the LFP signals and energy within.

**Signal recovery.** To further explain the above results, we visualize the recovered coefficients. As shown in Fig. 5, our reconstructed strong signal (marked by blue) can remove the bias. Meanwhile, the estimated $\theta(t)$ (marked by red) can capture the information of the weak signal.

## 7 CONCLUSION

In this paper, we propose the SWDI for neural signal recovery, which can simultaneously remove bias in estimating the strong signal and capture weak components in estimating the whole signal. Our method has a unique closed-form solution and can achieve better estimations than Wavelet Shrinkage. Besides, we provide an efficient discretization algorithm that can efficiently obtain the whole solution path. In PD analysis, our method identifies the non-burst/tonic activity in the low $\beta$ band, which has been recently found to be responsive to medical treatment.

**Limitations and future work.** We only discuss the signal recovery in a non-adaptive way. However, the sub-band and spatially adaptive wavelet decomposition can achieve better reconstruction results. While saying so, we have shown promising results in real applications. Moreover, by considering the strong and the weak signals on each sub-band, our method can be potentially applied to adaptive decomposition and will be carefully investigated in the future.

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

APPENDIX

## A  RELATED WORK OF DIFFERENTIAL INCLUSION IN SIGNAL RECOVERY

The differential inclusion method in signal recovery was proposed in Osher et al. (2016), which is called (Linearized) *Bregman Inverse Scale Space* and can be viewed as a continuous dynamics of the *Linearized Bregman Iteration* (LBI) proposed in Osher et al. (2005); Yin et al. (2008) for image denoising. From the perspective of differential inclusion, Osher et al. (2016) firstly showed the model selection consistency property, recovering the true signal set under irrepresentable conditions. The Huang & Yao (2018) later generalized this result to from the linear model to a general convex function. Moreover, Huang et al. (2016; 2020) proposed the variable splitting method which leads to better model selection consistency. The Sun et al. (2017) further applied them to Alzheimer's Disease and found there exists another type of lesion feature (which they called "procedural bias") that can help disease diagnosis. Our method is motivated by differences from this method in the Wavelet Denoising scenario, in which our primary goal is to reconstruct the signal. Besides, thanks to the orthogonality of the Wavelet Matrix, our method has a closed-form solution, which leads to theoretical advantages over Wavelet Shrinkage.

## B  SUPPORTING LEMMAS

**Lemma B.1** (Concentration for Lipschitz functions). *Let $(X_1, \ldots, X_n)$ be a vector of i.i.d. standard Gaussian variables, and let $f : \mathbb{R}^n \to \mathbb{R}$ be L-Lipschitz with respect to the Euclidean norm. Then the variable $f(X) - \mathbb{E}[f(X)]$ is sub-Gaussian with parameter at most L, and hence*

$$\mathbb{P}[|f(X) - \mathbb{E}[f(X)]| \geq t] \leq 2e^{-\frac{t^2}{2L^2}} \quad \text{for all } t \geq 0.$$

**Lemma B.2** (Hoeffding bound). *Suppose that the variables $X_i, i = 1, \ldots, n$ are independent, and $X_i$ has mean $\mu_i$ and sub-Gaussian parameter $\sigma_i$. Then for all $t \geq 0$, we have*

$$\mathbb{P}\left[\sum_{i=1}^{n}(X_i - \mu_i) \geq t\right] \leq \exp\left\{-\frac{t^2}{2\sum_{i=1}^{n}\sigma_i^2}\right\}.$$

**Lemma B.3** ($\chi^2$-variables). *Let $(X_1, \ldots, X_n)$ be a vector of i.i.d. standard Gaussian variables. Then $\left(X_1^2, \ldots, X_n^2\right)$ are i.i.d Chi-squared variables with 1 degree of freedom. Then we have*

$$\mathbb{P}\left[\frac{1}{n}\left|\sum_{i=1}^{n}X_i^2 - n\right| \geq t\right] \leq e^{-\frac{nt^2}{8}}, \quad \forall t \in (0, 1).$$

**Lemma B.4** (Expectation of Maximum Gaussian (Theorem 1 in Kamath (2015))). *Let Let $(X_1, \ldots, X_n)$ be a vector of i.i.d. standard Gaussian variables, then*

$$\mathbb{E}[\max_{1 \leq i \leq n} X_i] \leq \sqrt{2 \log n}.$$

$$|f(Ax) - f(Ay)| \leq \sqrt{\max_{i=1,\ldots,n}(A^T A)_{ii}}\|x - y\|_2, \quad x, y \in \mathbb{R}^n.$$

## C  PROOF OF SECTION 3

**Proposition C.1.** *We have $0 = \arg\inf_\lambda \sup_{|\theta_i^*| \geq 1} \mathbb{E}(\eta(\omega_i, \lambda) - \theta_i^*)^2$.*

*Proof.* Since $\theta_i^* \geq 1$, then

$$\arg\inf_\lambda \sup_{\theta_i^* \geq 1} \frac{\mathbb{E}(\eta(\omega_i, \lambda) - \theta_i^*)^2}{n^{-1} + \min((\theta_i^*)^2, 1)} = \arg\inf_\lambda \sup_{\theta_i^* \geq 1} \mathbb{E}(\eta(\omega_i, \lambda) - \theta_i^*)^2$$

According to Theorem 2 in Donoho & Johnstone (1994), we have

$$\mathbb{E}(\eta(\omega_i, \lambda) - \theta_i^*)^2 = 1 + \lambda^2 + \left((\theta_i^*)^2 - \lambda^2 - 1\right)\left\{\Phi(\lambda - \theta_i^*) - \Phi(-\lambda - \theta_i^*)\right\}$$
$$- (\lambda - \theta_i^*)\phi(\lambda + \theta_i^*) - (\lambda + \theta_i^*)\phi(\lambda - \theta_i^*).$$

Denote $g(\lambda, \theta_i *) := \mathbb{E}(\eta(\omega_i, \lambda) - \theta_i^*)^2$. Then it is sufficient to show that for each $\theta_i^*$, we have

$$g(\lambda, \theta_i *) \geq g(\lambda, 0) \geq g(0, 0) = g(0, \theta_i *). \tag{7}$$

According to Lemma 1 in Donoho & Johnstone (1994), we have $g(\lambda, \theta_i *)$ is increasing w.r.t. $|\theta_i^*|$, therefore we have $g(\lambda, \theta_i *) \geq g(\lambda, 0)$. The "=" in Eq. 7 is also obvious since

$$g(0, \theta_i *) = 1, \forall \, \theta_i * .$$

It is left to prove the 2nd "$\geq$" in Eq. 7. It is suffcient to show that

$$g(\lambda, 0) = (1 + \lambda^2)(1 - \Phi(\lambda) + \Phi(-\lambda)) - 2\lambda\phi(\lambda)$$

is non-increasing w.r.t. $\lambda$. Take the gradient of $g(\lambda, 0)$ w.r.t. $\lambda$, we have

$$\frac{\partial g(\lambda, 0)}{\partial \lambda} = -2(1 + \lambda^2)\phi(\lambda) + 2\lambda(1 - \Phi(\lambda) + \Phi(-\lambda)) - 2\phi(\lambda) + 2\lambda^2\phi(\lambda)$$

$$= -4\phi(\lambda) + 2\lambda(1 - \Phi(\lambda) + \Phi(-\lambda)).$$

Then it is sufficient to show that $\frac{\partial g(\lambda, 0)}{\partial \lambda} \leq 0$. To show this, we first consider the 2nd derivative of $g(\lambda, 0)$, which gives

$$\frac{\partial^2 g(\lambda, 0)}{\partial \lambda^2} = 4\lambda\phi(\lambda) + 2(1 - \Phi(\lambda) + \Phi(-\lambda)) + 2\lambda(-2\phi(\lambda))$$

$$= 2(1 - \Phi(\lambda) + \Phi(-\lambda)) \geq 0.$$

This means $l(\lambda) := \frac{\partial g(\lambda, 0)}{\partial \lambda}$ is non-decreasing. We consider $\lim_{\lambda \to \infty} l(\lambda)$, according to the Hôpital's rule, we have

$$\lim_{\lambda \to \infty} l(\lambda) = \frac{\lim_{\lambda \to \infty} 2(1 - \Phi(\lambda) + \Phi(-\lambda))}{\lim_{\lambda \to \infty} \frac{1}{\lambda}} = \frac{\lim_{\lambda \to \infty} 4\phi(\lambda)}{\lim_{\lambda \to \infty} \frac{1}{\lambda^2}} = \lim_{\lambda \to \infty} 4\lambda^2\phi(\lambda) = 0.$$

This means $l(\lambda) := \frac{\partial g(\lambda, 0)}{\partial \lambda} \leq 0$. Therefore, we have $g(\lambda, 0) \geq g(0, 0)$. Hence,

$$\arg\inf_\lambda \sup_{\theta_i^* \geq 1} \mathbb{E}(\eta(\omega_i, \lambda) - \theta_i^*)^2 = 0.$$

The proof is finished. $\qquad\square$

## D   PROOF OF SECTION 4.1

**Proposition D.1.** *The solution of differential inclusion in Eq. 1 is $\theta_j(t) = \omega_j(t)$ for $t \geq \frac{1}{|\omega_j|}$; and $= 0$ otherwise for each $j$.*

*Proof.* Note that $\rho(0) = \theta(0) = 0$. We define $t_j := \sup_t\{\rho_j(0) + t\omega_j \in \partial|\theta_j(0)|\}$. Then we define $\theta_j(t) = 0$ for $t < t_j$ and $= \omega_j$ for $t \geq t_j$ and $\rho_j(t) = \omega_j t$ for $t < t_j$ and $= \text{sign}(\omega_j)$ for $t \geq t_j$. It can be shown that this defined $\{\rho(t), \theta(t)\}$ is the solution of Eq. 1. This solution is unique since the loss $\ell(\theta(t)) := \frac{1}{2}\|\omega - \theta(t)\|_2^2$ is strictly convex w.r.t. $\theta$. According to Theorem 2.1 in Osher et al. (2016), we know that the $\theta(t)$ is unique. $\qquad\square$

**Theorem D.2.** *Denote $\theta_{\min}^{*,s} := \min_{i \in S}|\theta_i^*|$. Then at $\bar{\tau} := \frac{1}{(1+a)\sqrt{2\log n}}$ for some $a > 0$ and $\theta_{\min}^{*,s} \geq 2(1+a)\sqrt{2\log n}$ and $\frac{a}{2}\sqrt{2\log n} > \theta_j^*$ for $j \notin S$. Then with probability at least $1 - \frac{2}{n^{4a^2}} - \max\left(\exp\left(-\frac{s\lambda^2}{8}\right), \frac{1}{n^{\frac{(1+a)^2}{4}}}\right)$, we have*

$$\|\theta(\bar{\tau}) - \theta^{*,s}\|_2 < \|\eta(\omega, \lambda) - \theta^{*,s}\|_2, \tag{8}$$

*where $\theta(t)$ is the solution of Eq. 1.*

*Proof.* First we show the model selection consistency: $\text{supp}(\theta(\bar{\tau})) = S$. According to Prop. D.1, it is sufficient to show that $\bar{\tau} \geq \frac{1}{|\omega_j := \theta_j^{*,s} + \varepsilon_j|}$ for all $j \in S$ and $\bar{\tau} < \frac{1}{|\omega_j := \theta_j^* + \varepsilon_j|}$ for all $j \notin S$. Since $|\theta_j^*| \leq \frac{a}{2}\sqrt{2\log n}$, it is sufficient to show that $\max_{1 \leq j \leq n}|\varepsilon_j| \leq (1 + \frac{a}{2})\sqrt{2\log n}$ with high

probability, which can ensure that $|\theta_j^{*,s} + \varepsilon_j| \geq \theta_{\min}^{*,s} - \max_{1 \leq j \leq n} |\varepsilon_j| > (1+a)\sqrt{2 \log n} = \frac{1}{\tau}$ for $j \in S$ and $|\theta_j^* + \varepsilon_j| < \frac{a}{2}\sqrt{2 \log n} + \max_{1 \leq j \leq n} |\varepsilon_j| \leq (1+a)\sqrt{2 \log n} = \frac{1}{\tau}$. Since $\max_{1 \leq j \leq n} |\varepsilon_j| < \max\{\max_{1 \leq n} \varepsilon_j, \max_{1 \leq n} -\varepsilon_j\}$, then we have

$$P(\max_{1 \leq j \leq n} |\varepsilon_j| > (1 + \frac{a}{2})\sqrt{2 \log n})$$

$$\leq P\left(\max_{1 \leq j \leq n} \varepsilon_j > (1 + \frac{a}{2})\sqrt{2 \log n}\right) + P\left(\max_{1 \leq j \leq n} -\varepsilon_j > (1 + \frac{a}{2})\sqrt{2 \log n}\right)$$

$$\leq 2P\left(\max_{1 \leq j \leq n} \varepsilon_j > (1 + \frac{a}{2})\sqrt{2 \log n}\right) = 2P\left(\max_{1 \leq j \leq n} \varepsilon_j - \sqrt{2 \log n} > \frac{a}{2}\sqrt{2 \log n}\right)$$

$$\leq 2P\left(\max_{1 \leq j \leq n} \varepsilon_j - \mathbb{E}\left[\max_{1 \leq j \leq n} \varepsilon_j\right] > \frac{a}{2}\sqrt{2 \log n}\right).$$

According to lemma **??** and lemma B.1, we have

$$P\left(\max_{1 \leq j \leq n} \varepsilon_j - \mathbb{E}\left[\max_{1 \leq j \leq n} \varepsilon_j\right] > \frac{a}{2}\sqrt{2 \log n)}\right) < \frac{1}{n^{\frac{a^2}{4}}}.$$

To prove Eq. 25, without loss of generality we assume that $\theta_j^{*,s} > 0$ for $j \in S$. Then we have

$$\|\theta(\bar{\tau}) - \theta^{*,s}\|_2^2 = \|\varepsilon_S\|_2^2 \tag{9}$$

and that

$$\left(\eta(\omega, \lambda)_j - \theta_j^{*,s}\right)^2 = \begin{cases} (\theta_j^{*,s})^2, & \lambda \geq \theta_j^{*,s} + \varepsilon_j \\ (\varepsilon_j - \lambda)^2, & \lambda < \theta_j^{*,s} + \varepsilon_j \end{cases}$$

We denote $\mathrm{I} := \{j : \lambda \geq \theta_j^{*,s} + \varepsilon_j\}$ and $\mathrm{II} = S - \mathrm{I}$. Then we have

$$\|\eta(\omega, \lambda) - \theta^{*,s}\|_2^2 \geq \|\eta(\omega, \lambda)_S - \theta^{*,s}\|_2^2 = \|\theta_{\mathrm{I}}^{*,s}\|_2^2 + \|\varepsilon_{\mathrm{II}}\|_2^2 - 2\lambda \sum_{j \in \mathrm{II}} \varepsilon_j + \lambda^2(s - |\mathrm{I}|) \tag{10}$$

If $\lambda < (1+a)\sqrt{2 \log n}$, then condition on $\max_{1 \leq j \leq n} |\varepsilon_j| \leq (1 + \frac{a}{2})\sqrt{2 \log n}$, we have $\mathrm{I} = \emptyset$. Combining Eq. 9 and Eq. 10, it is sufficient to show that

$$\sum_{j \in S} \varepsilon_j < \frac{s\lambda}{2},$$

which has probability at least $1 - \exp\left(\frac{-s\lambda^2}{8}\right)$ applying Lemma B.2. Otherwise, if $\lambda \geq (1+a)\sqrt{2 \log n}$, then condition on $\max_{1 \leq j \leq n} |\varepsilon_j| \leq (1 + \frac{a}{2})\sqrt{2 \log n}$, we have $\|\varepsilon_{\mathrm{I}}\|_2^2 < \|\theta_{\mathrm{I}}^{*,s}\|_2^2$. Combining Eq. 9 and Eq. 10, it is sufficient to show that

$$\sum_{j \in \mathrm{II}} \varepsilon_j < \frac{|\mathrm{II}|\lambda}{2},$$

which has probability at least $1 - \frac{1}{n^{\frac{(1+a)^2}{4}}}$ applying Lemma B.2. The proof is completed. $\square$

## D.1 LINEARIZED WAVELET DIFFERENTIAL INCLUSION

Similar to SWDI, we also provide a linearized version in the following:

$$\dot{\rho}(t) + \frac{\dot{\theta}(t)}{\kappa} = \omega - \theta(t), \tag{11a}$$

$$\rho(t) \in \partial \|\theta(t)\|_1. \tag{11b}$$

**Proposition D.3.** *The solution of differential inclusion in Eq. 11 is*

$$\theta_j(t) = \begin{cases} \omega_j \left(1 - \exp\left(-\kappa\left(t - \frac{1}{\omega_j}\right)\right)\right), & t \geq \frac{1}{|\omega_j|} \\ 0, & otherwise \end{cases} \quad \forall j.$$

*Proof.* Similar to Prop. D.1, we define $t_j$ for each $j$ and $\rho_j(t) = \omega_j t$ for $t < t_j$ and $= \text{sign}(\theta_j)$ for $t \geq t_j$. It is easy to validate that such defined $(\rho(t), \theta(t))$ is the solution of Eq. 11. To see the uniqueness, we denote $v(t) := \rho(t) + \frac{\theta(t)}{\kappa}$, then we have

$$\dot{z}(t) = \omega - \kappa\eta(z(t), 1) := g(z(t)).$$

Since $g(\cdot)$ is Lipschitz continuous, the solution is unique according to the Picard-Lindelöf Theorem. $\square$

**Theorem D.4.** *Under the same conditions and the definition of $\bar{\tau}$ in Thm. D.2. Then with probability at least $1 - \frac{2}{n^{4a^2}} - \max\left(\exp\left(-\frac{s\lambda^2}{32}\right), \frac{1}{n^{\frac{(1+a)^2}{16}}}\right)$, we have*

$$\|\theta(\bar{\tau}) - \theta^{*,s}\|_2 < \|\eta(\omega, \lambda) - \theta^{*,s}\|_2, \tag{12}$$

*where $\theta(t)$ is the solution of Eq. 11.*

*Proof.* Similar to Thm. D.2, conditioning on $\max_{1 \leq j \leq n} |\varepsilon_j| \leq (1 + \frac{a}{2})\sqrt{2\log n}$ (with probability at least $1 - \frac{2}{n^{4a^2}}$), we have that $\text{supp}(\theta(\bar{\tau})) = S$, by additionally noting that $\bar{\tau}$ is exactly greater than $\frac{1}{\omega_j}$ for any $j \in S$. Next, we show that with probability at least $1 - \max\left(\exp\left(-\frac{s\lambda^2}{32}\right), \frac{1}{n^{\frac{(1+a)^2}{16}}}\right)$, we have

$$\|\bar{\theta}(\bar{\tau}) - \theta^{*,s}\|_2^2 < \|\eta(\omega, \lambda) - \theta^{*,s}\|_2^2 - \min\left(\frac{\lambda^2}{2}, (1+a)^2 \log n\right), \tag{13}$$

where $\bar{\theta}(t) = \lim_{\kappa \to \infty} \theta_\kappa(t)$ with $\theta_\kappa(t)$ being the solution of Eq. 11 with a fixed $\kappa > 0$. According to Prop. D.1, D.3, the $\bar{\theta}(t)$ is the solution of Eq. 11. If Eq. 13 holds, then due to the continuity of $\theta_\kappa(t)$ with respect to $\kappa$, we have Eq. 12 as long as $\kappa$ is large enough. Similar to the proof for Thm. D.2, when $\lambda < (1+a)\sqrt{2\log n}$, conditioning on $\max_{1 \leq j \leq n} |\varepsilon_j| \leq (1 + \frac{a}{2})\sqrt{2\log n}$ we have $\sum_{j \in S} \varepsilon_j < \frac{s\lambda}{4}$ and hence

$$\|\bar{\theta}(\bar{\tau}) - \theta^{*,s}\|_2^2 < \|\eta(\omega, \lambda) - \theta^{*,s}\|_2^2 - \frac{\lambda^2}{2}$$

with probability at least $1 - \exp\left(\frac{-s\lambda^2}{32}\right)$. When $\lambda \geq (1+a)\sqrt{2\log n}$, we have $\sum_{j \in \text{II}} \varepsilon_j < \frac{|\text{II}|\lambda}{2}$ and therefore

$$\|\bar{\theta}(\bar{\tau}) - \theta^{*,s}\|_2^2 < \|\eta(\omega, \lambda) - \theta^{*,s}\|_2^2 - (1+a)^2 \log n$$

with probability at least $1 - \frac{1}{n^{\frac{(1+a)^2}{16}}}$. $\square$

# E    PROOF OF SECTION 4.2

**Proposition E.1.** *The solution of differential inclusion in Eq. 3 is*

$$\begin{cases} \theta_j(t) = \gamma_j(t) = \omega_j, & t \geq \frac{1 + \frac{1}{\beta}}{\omega_j} \\ \theta_j(t) = \frac{\omega_j}{1+\beta}, \ \gamma_j(t) = 0 & t < \frac{1 + \frac{1}{\beta}}{\omega_j} \end{cases} \forall j.$$

*Proof.* Note that $\gamma(0) = \rho(0) = 0$. Then we define $t_j$ for each $j$ as

$$t_j := \sup_{t > 0}\{\rho_j(0) + \frac{t\omega_j}{1+\beta} \notin \partial|\gamma_j(0)|\}.$$

When $t < t_j$, we have $|\rho_j(t)| < 1$ thus $\gamma_j(t) = 0$ and also $\theta_j = \frac{\omega_j}{1+\beta}$ according to Eq. 3a. For $t \geq t_j$, we have $|\rho_j(t)| = 1$ and thus $\dot{\rho}_j(t) = 0$. Therefore, we have $\gamma_j(t) = \theta_j(t) = \omega_j$. It is easy to see such defined $(\rho(t), \theta(t), \gamma(t))$ is the solution of Eq. 3. We can obtain that $t_j = \frac{1 + \frac{1}{\beta}}{\omega_j}$. According to Huang et al. (2016), this solution is unique. $\square$

**Theorem E.2.** *Denote $\theta^*_{\max,T} := \max_{i \in T} |\theta^*_i|$ and $n$ is large enough such that $\theta^*_{\max,T} < a_0 \sqrt{\log n}$ for some $0 < a_0 < 1$. Then for $(\theta(t), \theta^s(t))$ in Eq. 3, if $n > 4^{1/(1-a_0)}$ at $\bar{\tau} := \frac{1 + \frac{1}{\rho}}{(1+a)\sqrt{2 \log n}}$, the following holds with probability at least*

$$
1 - \frac{2}{n^{4a^2}} - \max\left(\frac{1}{n^{\frac{s}{32}}}, \frac{1}{n^{\frac{(1+a)^2}{16}}}\right) - \exp\left(-\frac{\sum_{i \in T}(\theta^*_i)^2}{72}\right) - \exp\left(\frac{-n^{1-a_0}\sum_{i \in T}(\theta^*_i)^2}{24(2 + \log n)}\right)
$$

$$
- \exp\left(-\frac{|T|\max\left(1, \frac{\sum_{i \in T}(\theta^*_i)^2}{|T|} - 1\right)}{8}\right) - \exp\left(-\frac{|N|\max\left(1, \frac{\sum_{i \in T}(\theta^*_i)^2}{|T|} - 1\right)}{8}\right):
$$

1. **Strong Signal Recovery.** *For the strong signal coefficients $\theta^{*,s}_S$,*

$$
\|\theta^s(\bar{\tau}) - \theta^{*,S}\|_2 = \|\theta_S(\bar{\tau}) - \theta^{*,s}_S\|_2 < \|\eta(\omega, \lambda) - \theta^{*,s}\|_2, \ \forall \lambda > 0. \tag{14}
$$

2. **Weak Signal Recovery.** *For the weak signal coefficients $\theta^*_T$, there exists $\infty > \rho^* > 0$ such that*

$$
\|\theta(\bar{\tau})_T - \theta^*_T\|_2 < \|0 - \theta^*_T\|_2 = \|\theta^*_T\|_2 \ i.e., \ \rho = \infty \ \textit{Shrinkage to 0};
$$
$$
\|\theta(\bar{\tau})_T - \theta^*_T\|_2 < \|\omega_T - \theta^*_T\|_2 = \|\varepsilon_T\|_2 \ i.e., \ \rho = 0 \ \textit{No Shrinkage}.
$$

   *We can obtain a similar result for $\theta^*_{S^c}$, where $S^c := T \cup N$ contains the weak and null components.*

3. **Whole Signal Recovery.** *For $\theta^*$, under the same $\rho^*$ in item 2, we have*

$$
\|\theta(\bar{\tau}) - \theta^*\|_2 < \|\eta(\omega, \lambda) - \theta^*\|_2, \ \forall \lambda \geq \sqrt{\log n}.
$$

*Proof.* The proof of **Strong Signal Recovery** is the same as the proof for Thm. D.2. For **Weak Signal Recovery**, we define $I(p) = \|(1 - p)\omega_T - \theta^*_T\|_2^2 = \|p\theta^*_T - (1 - p)\varepsilon_T\|_2^2$. We then have

$$
I'(p) = 2p\sum_{i \in T}(\theta^*_i + \varepsilon_i)^2 - 2\sum_{i \in T}\varepsilon_i(\theta^*_i + \varepsilon_i).
$$

We obtain the minimizer of $I(p)$ as $p^* = \frac{\sum_{i \in T}\varepsilon_i^2 + \sum_{i \in T}\varepsilon_i\theta^*_i}{\sum_{i \in T}(\theta^*_i + \varepsilon_i)^2}$ by setting $I(p') = 0$. If we can show that $0 < p^* < 1$, then there exists a $\beta^*$ such that $p^* = \frac{\beta^*}{1 + \beta^*}$. Then it is sufficient to show that

$$
\sum_i \varepsilon_i\theta^*_i + \sum_i \varepsilon_i^2 > 0, \tag{15}
$$

$$
\sum_i \varepsilon_i\theta^*_i + \frac{1}{6}\sum_i \theta^*_i > 0. \tag{16}
$$

For Eq. 15, we have that:

$$
P\left(\sum_i \varepsilon_i\theta^*_i + \sum_i \varepsilon_i^2 < 0\right)
$$

$$
= P\left(\sum_i \varepsilon_i\theta^*_i + \sum_i \varepsilon_i^2 < 0, \sum_i \varepsilon_i^2 < \frac{|T|}{2}\right) + P\left(\sum_i \varepsilon_i\theta^*_i + \sum_i \varepsilon_i^2 < 0, \sum_i \varepsilon_i^2 \geq \frac{|T|}{2}\right)
$$

$$
\leq P\left(\sum_i \varepsilon_i^2 < \frac{|T|}{2}\right) + P\left(\sum_i \varepsilon_i\theta^*_i < -\frac{|T|}{2}\right)
$$

Applying Lemma B.3 to the first term and Lemma B.2 to the second term, we have that

$$
P\left(\sum_i \varepsilon_i^2 < \frac{|T|}{2}\right) < \exp\left(-\frac{|T|}{8}\right), \ P\left(\sum_i \varepsilon_i\theta^*_i < -\frac{|T|}{2}\right) < \exp\left(-\frac{|T|^2}{8\sum_{i \in T}(\theta^*_i)^2}\right).
$$

For Eq. 16, we have

$$
P\left(\sum_i \varepsilon_i\theta^*_i + \frac{1}{4}\sum_i(\theta^*_i)^2 < 0\right) = P\left(\sum_i \varepsilon_i\theta^*_i < -\frac{\sum_i(\theta^*_i)^2}{6}\right).
$$

Applying Lemma B.2, we have that

$$P\left(\sum_i \varepsilon_i \theta_i^* < -\frac{\sum_{i \in T}(\theta_i^*)^2}{4}\right) \leq \exp\left(-\frac{\sum_{i \in T}(\theta_i^*)^2}{72}\right).$$

Since $\|\theta_T^*\|_2$ is the $\ell_2$ error of $0_T$ with $p = 1$ and $\|\varepsilon_T\|_2$ is the $\ell_2$ error of $\omega_T$ with $p = 0$, we obtain the conclusion. To extend this result to $\theta_{S^c}^*$, following the same procedure, we can obtain that

$$p^* = \frac{\sum_{i \in T} \varepsilon_i^2 + \sum_{i \in T} \varepsilon_i \theta_i^* + \sum_{j \in N} \varepsilon_j^2}{\sum_{i \in T}(\theta_i^* + \varepsilon_i)^2 + \sum_{j \in N} \varepsilon_j^2},$$

which is $< 1$ and $> 0$ if Eq. 15, 16 holds. Finally, we prove **the Whole Signal Recovery**. According to the above results, it is sufficient to show that there exists a $\beta$ such that

$$\|\theta(\bar{\tau})_{S^c} - \theta_{S^c}^*\|_2 < \|\eta(\omega, \lambda)_T - \theta_T^*\|_2 < \|\eta(\omega, \lambda)_{S^c} - \theta_{S^c}^*\|_2.$$

condition on $\|\theta(\bar{\tau})_S - \theta_S^*\|_2 < \|\eta(\omega, \lambda)_S - \theta_S^*\|_2$ that holds with high probability. We first show that

$$\|\eta(\omega, \lambda)_T - \theta_T^*\|_2^2 > \frac{1}{3}\|\theta_T^*\|_2^2, \tag{17}$$

$$\|\theta(\bar{\tau})_{S^c} - \theta_{S^c}^*\|_2^2 < \frac{2}{3}\|\theta_T^*\|_2^2, \tag{18}$$

for some $\infty > \beta > 0$. For Eq. 17, we denote $b_i := \eta(\omega, \lambda)_i$, then we have

$$\|\eta(\omega, \lambda)_T - \theta_T^*\|_2^2 = \sum_{i \in T} b_i^2 + \sum_{i \in T} 2\theta_i^* b_i + \sum_{i \in T}(\theta_i^*)^2 \geq \sum_{i \in T} 2\theta_i^* b_i + \sum_{i \in T}(\theta_i^*)^2,$$

which holds as long as $\sum_{i \in T} \theta_i^* b_i \geq -\frac{1}{6}\sum_{i \in T}(\theta_i^*)^2$. We then have

$$P\left(\sum_{i \in T} \theta_i^* b_i \leq -\frac{1}{6}\sum_{i \in T}(\theta_i^*)^2\right) = P\left(\sum_{i \in T} \theta_i^* b_i - \sum_{i \in T} \theta_i^* \mathbb{E}[b_i] \leq -\frac{1}{6}\sum_{i \in T}(\theta_i^*)^2 - \sum_{i \in T} \theta_i^* \mathbb{E}[b_i]\right)$$

$$\leq \exp\left(-\frac{\left(\frac{1}{6}\sum_{i \in T}(\theta_i^*)^2 + \sum_{i \in T} \theta_i^* \mathbb{E}[b_i]\right)^2}{2\sum_{i \in T}(\theta_i^*)^2 \mathbb{E}[b_i^2]}\right). \tag{19}$$

Without loss of generality, we assume $\theta_i^* > 0$. Then for $\mathbb{E}[b_i]$, we have

$$\mathbb{E}[b_i] = \frac{1}{\sqrt{2\pi}}\left(\int_{\sqrt{\log n} - \theta_i^*}^{+\infty} x \exp\left(-\frac{x^2}{2}\right)dx + \int_{-\infty}^{-\sqrt{\log n} - \theta_i^*} x \exp\left(-\frac{x^2}{2}\right)dx\right)$$

$$= -\frac{1}{\sqrt{2\pi}}\int_{-\sqrt{\log n} - \theta_i^*}^{-\sqrt{\log n} + \theta_i^*} x \exp\left(-\frac{x^2}{2}\right)dx$$

$$> -\frac{1}{\sqrt{2\pi}}\frac{1}{n^{1 - \frac{\theta_i^*}{\sqrt{\log n}}}} \geq -\frac{1}{\sqrt{2\pi}}\frac{1}{n^{1 - a_0}}.$$

For $\mathbb{E}[b_i^2]$, we have

$$\mathbb{E}[b_i^2] = \frac{1}{\sqrt{2\pi}}\left(\underbrace{\int_{\sqrt{\log n} - \theta_i^*}^{+\infty} x \exp\left(-\frac{x^2}{2}\right)dx}_{J_1} + \underbrace{\int_{-\infty}^{-\sqrt{\log n} - \theta_i^*} x \exp\left(-\frac{x^2}{2}\right)dx}_{J_2}\right).$$

For $J_1$, we have

$$\frac{1}{\sqrt{2\pi}}J_1 = \frac{1}{\sqrt{2\pi}}\int_{\sqrt{\log n} - \theta_i^*}^{+\infty} x d\left(-\exp\left(-\frac{x^2}{2}\right)\right)$$

$$= \frac{1}{\sqrt{2\pi}} - x \exp\left(-\frac{x^2}{2}\right)\Big|_{\sqrt{\log n} - \theta_i^*}^{+\infty} + P(\varepsilon_i > \sqrt{\log n} - \theta_i^*)$$

$$= P(\varepsilon_i > \sqrt{\log n} - \theta_i^*) + \frac{1}{\sqrt{2\pi}}\left(\sqrt{\log n} - \theta_i^*\right)\exp\left(-\frac{\left(\sqrt{\log n} - \theta_i^*\right)^2}{2}\right)$$

Similarly, for $J_2$, we have

$$\frac{1}{\sqrt{2\pi}}J_2 = P(\varepsilon_i > \sqrt{\log n} + \theta_i^*) + \frac{1}{\sqrt{2\pi}}\left(\sqrt{\log n} + \theta_i^*\right)\exp\left(-\frac{\left(\sqrt{\log n} + \theta_i^*\right)^2}{2}\right)$$

$$\leq P(\varepsilon_i > \sqrt{\log n} - \theta_i^*) + \frac{1}{\sqrt{2\pi}}\left(\sqrt{\log n} + \theta_i^*\right)\exp\left(-\frac{\left(\sqrt{\log n} - \theta_i^*\right)^2}{2}\right).$$

Therefore, we have

$$\mathbb{E}[b_i^2] \leq 2P(\varepsilon_i > \sqrt{\log n} - \theta_i^*) + \frac{2}{\sqrt{2\pi}}\sqrt{\log n}\exp\left(-\frac{\left(\sqrt{\log n} - \theta_i^*\right)^2}{2}\right) \leq \frac{2 + \log n}{n^{1-a_0}}.$$

Substituting these results into Eq. 19, we have:

$$P\left(\sum_{i\in T}\theta_i^* b_i \leq -\frac{1}{6}\sum_{i\in T}(\theta_i^*)^2\right) \leq \exp\left(\frac{-n^{1-a_0}\left(\frac{1}{6} - \frac{1}{\sqrt{2\pi}n^{1-a_0}}\right)\sum_{i\in T}(\theta_i^*)^2}{2(2 + \log n)}\right),$$

$$\leq \exp\left(\frac{-n^{1-a_0}\sum_{i\in T}(\theta_i^*)^2}{24(2 + \log n)}\right),$$

as long as $n > 4^{1/(1-a_0)}$. Next we prove Eq. 18, which is equivalent to showing that

$$\sum_{i\in T}\varepsilon_i^2 + \sum_{j\in N}\varepsilon_j^2 - 2\beta\sum_{i\in T}\theta_i^*\varepsilon_i < \frac{2(2\beta + 1)}{3}\sum_{i\in T}(\theta_i^*)^2.$$

If we take $\beta = n/|T|$, then it is sufficient to show that

$$\frac{\sum_{i\in T}\varepsilon_i^2}{|T|} \leq \frac{1}{|T|}\sum_{i\in T}(\theta_i^*)^2, \tag{20}$$

$$\frac{\sum_{j\in N}\varepsilon_j^2}{|N|} \leq \frac{1}{|T|}\sum_{j\in T}(\theta_j^*)^2, \tag{21}$$

$$\sum_{i\in T}\varepsilon_i\theta_i^* + \frac{1}{6}\sum_{j\in T}(\theta_i^*)^2 > 0. \tag{22}$$

Conditioning on $\sum_i \varepsilon_i\theta_i^* + \frac{1}{6}\sum_i(\theta_i^*)^2 > 0$, $\sum_{i\in T}\varepsilon_i^2 \geq \frac{|T|}{2}$ and $\|\theta(\bar{\tau})_S - \theta_S^*\|_2 < \|\eta(\omega,\lambda)_S - \theta_S^*\|_2$, we have that Eq. 22 hold. For Eq. 20, we have

$$P\left(\frac{\sum_{i\in T}\varepsilon_i^2}{|T|} > \frac{1}{|T|}\sum_{i\in T}(\theta_i^*)^2\right) \leq \exp\left(-\frac{|T|\max\left(1, \frac{\sum_{i\in T}(\theta_i^*)^2}{|T|} - 1\right)}{8}\right).$$

Similarly, for Eq. 21, we have

$$P\left(\frac{\sum_{j\in N}\varepsilon_j^2}{|N|} > \frac{1}{|T|}\sum_{i\in T}(\theta_i^*)^2\right) \leq \exp\left(-\frac{|N|\max\left(1, \frac{\sum_{i\in T}(\theta_i^*)^2}{|T|} - 1\right)}{8}\right).$$

Summarizing these conclusions together, we have with probability at least

$$1 - \frac{2}{n^{4a^2}} - \max\left(\frac{1}{n^{\frac{s}{32}}}, \frac{1}{n^{\frac{(1+a)^2}{16}}}\right) - \exp\left(-\frac{\sum_{i\in T}(\theta_i^*)^2}{72}\right) - \exp\left(\frac{-n^{1-a_0}\sum_{i\in T}(\theta_i^*)^2}{24(2 + \log n)}\right)$$

$$- \exp\left(-\frac{|T|\max\left(1, \frac{\sum_{i\in T}(\theta_i^*)^2}{|T|} - 1\right)}{8}\right) - \exp\left(-\frac{|N|\max\left(1, \frac{\sum_{i\in T}(\theta_i^*)^2}{|T|} - 1\right)}{8}\right),$$

we have $\|\theta(\bar{\tau}) - \theta^*\|_2 < \|\eta(\omega,\lambda) - \theta^*\|_2$ for any $\lambda \geq \sqrt{\log n}$. $\qquad\square$

**Proposition E.3.** *Denote*

$$
t_j^* = \min_{t>0} \left\{ t : \int_0^t \frac{\omega(j)}{1 + \frac{1}{\beta}} \left( 1 - \exp\left( -\kappa\left(1+\beta\right)\left( \tilde{t} - \frac{1+\beta}{\omega(j)} \right) \right) \right) d\tilde{t} > 0 \right\}.
$$

*Then there exists a unique $\{C_1^1, ..., C_1^n\}$ with $C_1^j > 0$ and $\{C_2^1, ..., C_2^n\}$ with $C_2^j > 0$ such that*

- **Strong Signal Coefficients.** *For each $j$, when $t > t_j$,*

$$
\gamma_j(t) = C_1^j \exp\left( -\frac{\kappa t \left(1 + 2\beta - \sqrt{1+4\beta^2}\right)}{2} \right) + C_2^j \exp\left( -\frac{\kappa t \left(1 + 2\beta + \sqrt{1+4\beta^2}\right)}{2} \right) + \omega(j);
$$

$$
\tag{23a}
$$

$$
\theta_j(t) = \left( C_1^j + \frac{1}{\kappa\beta}\left( -\frac{\kappa t \left(1 + 2\beta - \sqrt{1+4\beta^2}\right)}{2} \right) \right) \exp\left( -\frac{\kappa t \left(1 + 2\beta - \sqrt{1+4\beta^2}\right)}{2} \right)
$$

$$
+ \left( C_2^j + \frac{1}{\kappa\beta}\left( -\frac{\kappa t \left(1 + 2\beta + \sqrt{1+4\beta^2}\right)}{2} \right) \right) \exp\left( -\frac{\kappa t \left(1 + 2\beta + \sqrt{1+4\beta^2}\right)}{2} \right) + \omega(j).
$$

$$
\tag{23b}
$$

- **Weak Signal Coefficients.** *For each $j$, when $t \le t_j$,*

$$
\theta_j(t) = \frac{\omega(j)}{1+\beta}\left( 1 - \exp\left( -\kappa\left(1+\beta\right)\left( t - \frac{1+\beta}{\omega(j)} \right) \right) \right) \text{ and } \gamma_j(t) = 0. \tag{24}
$$

*Proof.* It can be directly checked that that the Eq. 23a, 23b, 24 for any positive $\{C_1^1, ..., C_1^n\}$ and $\{C_2^1, ..., C_2^n\}$. To ensure the continuity of $\theta(t)$ and $\gamma(t)$ (they are continuous since they are differentiable) at $t_j$, we can determine $\{C_1^1, ..., C_1^n\}$ and $\{C_2^1, ..., C_2^n\}$. The uniqueness of $\{C_1^1, ..., C_1^n\}$ and $\{C_2^1, ..., C_2^n\}$ comes from the uniqueness of the solution $(\theta(t), \gamma(t))$, which can be similarly obtained by Picard-Lindelöf Theorem. □

**Theorem E.4.** *Under the same assumptions in Thm. E.2. Suppose $\kappa$ is sufficiently large, then we can obtain the same results in Thm. E.2 for Eq. 5.*

*Proof.* We can know from the solution of Eq. 5 given by Prop. E.3 and the solution of Eq. 3 given by Prop. E.1 that the solution of $(\theta(t), \gamma(t))$ in Eq. 23, 24 is continous with respect to $\kappa$ and converges to the solution of Eq. 3 given by Prop. E.1. Therefore the conclusions in Thm. E.2 hold when $\kappa$ is sufficiently large. □

# F    THEORETICAL ANALYSIS FOR DISCRETE FORM

**Proposition F.1.** *The $\ell_i(\theta(k), \gamma(k)) := \frac{1}{2}(\omega_i - \theta_i(k))^2 + \frac{\beta}{2}(\theta_i(k) - \gamma_i(k))^2$ is non-increasing as long as $\delta < \frac{2}{\kappa \max(1,\beta)}$ in Eq. 6b.*

*Proof.* Denote $H := \nabla^2 \ell_i(\theta(k), \gamma(k)) = \begin{pmatrix} 1+\beta & -\beta \\ -\beta & \beta \end{pmatrix}$. Denote $\begin{pmatrix} \theta_i(k+1) - \theta_i(k) \\ \gamma_i(k+1) - \gamma_i(k) \end{pmatrix} := \Delta_i(k)$. We have

$$\ell_i(\theta(k+1), \gamma(k+1)) - \ell_i(\theta(k), \gamma(k)) = \langle \nabla \ell_i(\theta(k), \gamma(k)), \Delta_i(k) \rangle + \frac{1}{2}\Delta_i(k)^\top H \Delta_k(k)$$

$$\leq -\frac{1}{\delta} \langle -\delta \nabla \ell(\theta(k), \gamma(k)), \Delta_i(k) \rangle + \frac{\|H\|_2}{2}\|\Delta_i(k)\|_2^2$$

$$\leq -\frac{1}{\delta} \left\langle \begin{pmatrix} \frac{\theta_i(k+1) - \theta_i(k)}{\kappa} \\ \rho_i(k+1) - \rho_i(k) + \frac{\gamma_i(k+1) - \gamma_i(k)}{\kappa} \end{pmatrix}, \Delta_i(k) \right\rangle + \frac{\|H\|_2}{2}\|\Delta_i(k)\|_2^2$$

$$\leq -\frac{1}{\delta} \langle \rho_i(k+1) - \rho_i(k), \gamma_i(k+1) - \gamma_i(k) \rangle + \left( \frac{\|H\|_2}{2} - \frac{1}{\kappa\delta} \right)\|\Delta_i(k)\|_2^2.$$

Since

$$\langle \rho_i(k+1) - \rho_i(k), \gamma_i(k+1) - \gamma_i(k) \rangle = |\gamma_i(k+1)| + |\gamma_i(k)|$$
$$- \rho_i(k) \cdot \gamma_i(k+1) - \rho_i(k+1) \cdot \gamma_i(k) \geq 0,$$

we have $\ell_i(\theta(k+1), \gamma(k+1)) \leq \ell(\theta(k), \gamma(k))$ as long as $\kappa\delta\|H\|_2 \leq 2$. Since $\|H\|_2 \leq \max(1, \beta)$, we have that $\delta < \frac{2}{\kappa \max(1,\beta)}$. $\square$

**Theorem F.2.** *Denote* $K := \lfloor \frac{(1+\frac{1}{\beta})\bar{\tau}}{\delta} \rfloor$ *with* $\delta = \frac{1}{\kappa(1+\beta)}$ *and* $\bar{\tau}$ *defined in Thm. D.2. Denote* $\theta_{\max}^* := \max_i |\theta_i^*|$. *Besides, we inherit the definition* $\theta_{\max,T}^*$ *in Thm. E.2. For* $(\theta(k), \gamma(k))$ *in Eq. 6, if* $n > 4^{1/(1-a_0)}$ *and* $\kappa$ *is sufficiently large, then with probability at least*

$$1 - \frac{2}{n^{4a^2}} - \max\left( \frac{1}{n^{\frac{s}{32}}}, \frac{1}{n^{\frac{(1+a)^2}{16}}} \right) - \exp\left( -\frac{\sum_{i \in T}(\theta_i^*)^2}{72} \right) - \exp\left( \frac{-n^{1-a_0}\sum_{i \in T}(\theta_i^*)^2}{24(2 + \log n)} \right)$$

$$- \exp\left( -\frac{|T|\max\left(1, \frac{\sum_{i \in T}(\theta_i^*)^2}{|T|} - 1\right)}{8} \right) - \exp\left( -\frac{|N|\max\left(1, \frac{\sum_{i \in T}(\theta_i^*)^2}{|T|} - 1\right)}{8} \right),$$

*we have*

1. **Strong Signal Recovery.** *For the strong signal coefficients* $\theta_S^{*,s}$,
$$\|\theta_S(K) - \theta_S^{*,s}\|_2 < \|\eta(\omega, \lambda)_S - \theta_S^{*,s}\|_2, \tag{25}$$
*for any* $\lambda > 0$.

2. **Weak Signal Recovery.** *For the weak signal coefficients and nulls* $\theta_T^*$, *there exists* $\infty > \beta^* > 0$ *such that*
$$\|\theta(K)_T - \theta_T^*\|_2 < \|\theta_T^*\|_2 \text{ i.e., } \beta = \infty \text{ Shrinkage to 0;}$$
$$\|\theta(K)_T - \theta_T^*\|_2 < \|\varepsilon_T\|_2 \text{ i.e., } \beta = 0 \text{ No Shrinkage.}$$
*We can obtain a similar result for* $\theta_{S^c}^*$, *the weak signal coefficients and null coefficients.*

3. **The Whole Signal Recovery.** *For the whole signal coefficients* $\theta^*$, *there exists* $\infty > \beta^* > 0$ *such that*
$$\|\theta(K) - \theta^*\|_2 < \|\eta(\omega, \lambda) - \theta^*\|_2$$
*for any* $\lambda \geq \sqrt{\log n}$. *That means Eq. 3 can be more accurate than Minimax and similar approaches in Donoho & Johnstone (1994).*

*Proof.* It is sufficient to prove that for any residue $e > 0$, there exists a $\kappa^o$ such that as long as $\kappa > \kappa^o$, the following condition holds:

$$\gamma_i(k) = 0, \forall k \leq K \text{ and } i \in S^c. \tag{26}$$

$$\left| \theta_i(K) - \frac{\omega_i}{1+\beta} \right| < e, \forall i. \tag{27}$$

$$|\theta_i(K) - w_i| < e, |\theta_i(K) - \gamma_i(K)| < e, \forall i \in S. \tag{28}$$

We first prove Eq. 26. With $z_i(k) := \rho_i(k) + \frac{\gamma_i(k)}{\kappa}$ and $z_i(0) = \theta_i(0) = 0$, it follows from Eq. 6 that

$$\frac{z_i(k+1) - z_i(k)}{\delta} = -\frac{\beta}{1+\beta}\left(\frac{\theta_i(k+1) - \theta_i(k)}{\kappa\delta} - \tilde{\varepsilon}_i\right).$$

where $\tilde{\varepsilon}_i := \theta_i^* + \varepsilon_i \ \forall i \in S^c$ and $\theta_i^* = 0$ if $i \in N$. Therefore, we have

$$z_i(k) = -\frac{\beta}{1+\beta}\frac{\theta_i(k)}{\kappa} + \frac{\beta}{1+\beta}k\delta\tilde{\varepsilon}_i.$$

Note that $\ell_i(k) := \ell_i(\theta(k), \gamma(k)) := \frac{1}{2}(\omega_i - \theta_i(k))^2 + \frac{\beta}{2}(\theta_i(k) - \gamma_i(k))^2$ in Eq. F.1 is non-increasing since $\delta < \frac{2}{\kappa\max(1,\beta)}$, therefore, we have

$$|\theta_i(k)| \le |\omega_i| \le \theta_{\max}^* + (1 + a/2)\sqrt{2\log n} \overset{\Delta}{=} B$$

condition on $\max_{1 \le j \le n}|\varepsilon_j| \le (1 + \frac{a}{2})\sqrt{2\log n}$. Then we have

$$|z_i(k)| \le \frac{\beta B}{(1+\beta)\kappa} + k\delta\frac{\beta}{1+\beta}\tilde{\varepsilon}_i < \frac{\beta B}{(1+\beta)\kappa} + \bar{\tau}(1+a)\sqrt{2\log n},$$

since $|\tilde{\varepsilon}_i| < \frac{a}{2}\sqrt{2\log n}$ and we have conditioned on $\max_{1 \le j \le n}|\varepsilon_j| \le (1 + \frac{a}{2})\sqrt{2\log n}$. Then there exists $\kappa^{(1)} > 0$ such that for any $\kappa > \kappa^{(1)}$, we have

$$|z_i(k)| < \frac{\beta B}{(1+\beta)\kappa} + \bar{\tau}(1+a)\sqrt{2\log n} < 1,$$

for any $k \le K$. This can prove Eq. 26. Next we prove Eq. 27. Note that Eq. 6b is equivalent to:

$$\theta_i(k+1) = \theta_i(k) - \kappa\delta\left((1+\beta)\theta_i(k) - \omega_i\right),$$

which implies

$$\theta_i(k+1) - \frac{\omega_i}{1+\beta} = (1 - \kappa\delta)\left(\theta_i(k) - \frac{\omega_i}{1+\beta}\right).$$

Denote $err_i(k) := \left|\theta_i(k) - \frac{\omega_i}{1+\beta}\right|$, we have $err_i(k) = (1 - \kappa\delta)^k err_i(0)$. Denote $K_e := \min\{k : err_i(k) < e,$ then we have

$$K_e \le \frac{\log e - \log err_i(0)}{\log(1 - \kappa\delta)} \le \frac{\log e - \log err_i(0)}{-\log 2}$$

$$\le \frac{\log e - \log B}{-\log 2} \ll \lfloor\frac{(1 + \frac{1}{\beta})\bar{\tau}}{\delta}\rfloor = \lfloor\kappa(1 + \frac{1}{\beta})\bar{\tau}\rfloor := K, \tag{29}$$

which will holds for any $\kappa > \kappa^{(2)}$ for some $\kappa^{(2)} > 0$. Finally we prove Eq. 28. Denote $K_{1,i} := \min\{k : |z_i(k)| \ge 1\}$ for $i \in S$. When $k < K_{1,i}$, we have

$$z_i(k) = \delta\beta\sum_{k=0}\theta_i(k).$$

Denote $K_{0,i} := \min\{|\theta_i(k)| > \frac{(1+3a/2)\sqrt{2\log n}}{1+\beta}$, where $|\theta_i(k)| > \frac{(1+3a/2)\sqrt{2\log n}}{1+\beta}$ can be implied by

$$|\theta_i(k) - \frac{\omega_i}{1+\beta}| < \frac{a/2\sqrt{2\log n}}{1+\beta} \overset{\Delta}{=} b$$

for $i \in S$ conditioning on $\max_{1 \le j \le n}|\varepsilon_j| \le (1 + \frac{a}{2})\sqrt{2\log n}$. According to Eq. 29, we have

$$K_{0,i}\delta \ll (1 + \frac{1}{\beta})\bar{\tau},$$

when $\kappa > \kappa^{(3)}$ for some $\kappa^{(3)} > 0$. Besides, it follows from the non-increasing property of $\ell_i(\theta(k), \gamma(k))$ that we have

$$\mathrm{sign}(\theta_i(k)) = \mathrm{sign}(\omega_i)$$

for any $k$ once $\theta_i(k) \neq 0$. If this does not hold, then at some $k$, we have

$$\ell_i(\theta(k), \gamma(k)) > \ell_i(\theta(0), \gamma(0)),$$

which contradicts to the non-increasing property. This means $\theta_i(k) = 0$ or does not change the sign once it becomes non-zero. Therefore, the $|z_i(k)|$ is non-decreasing and moreover, if $K_{1,i} > K_{0,i}$, then for any $k < K_{1,i}$

$$|z_i(k)| = \delta\beta \left| \left( \sum_{k=0}^{K_{0,i}} \theta_i(k) + \sum_{k=K_{0,i}+1} \theta_i(k) \right) \right| \geq \delta\beta \sum_{k=K_{0,i}+1} |\theta_i(k)|$$

$$\geq \delta(k - K_{0,i}) \frac{\beta(1 + 3a/2)\sqrt{2\log n}}{1 + \beta}.$$

Therefore, we have

$$\delta(K_{1,i} - K_{0,i}) \leq \frac{1 + \frac{1}{\beta}}{(1 + 3a/2)\sqrt{2\log n}} < \bar{\tau}.$$

Since $|z_i(k)|$ is non-decreasing, then once it is greater than 1, we have $z_i(k) = \gamma_i(k)$. Therefore, we have (recall that we define $\Delta_i(k) = \begin{pmatrix} \theta_i(k+1) - \theta_i(k) \\ \gamma_i(k+1) - \gamma_i(k) \end{pmatrix}$ in Prop. F.1)

$$\frac{\Delta_i(k)}{\kappa} = \delta \left( \begin{pmatrix} \omega_i \\ 0 \end{pmatrix} - H d_i(k) \right),$$

where $d_i(k) := \begin{pmatrix} \theta_i(k) \\ \gamma_i(k) \end{pmatrix}$ and $H$ is defined in Prop. F.1. Denote $\tilde{\delta}_i := \begin{pmatrix} \omega_i \\ 0 \end{pmatrix}$, then multiplying $H$ on both sides, we have

$$H d_i(k+1) - \tilde{\omega}_i = (I_2 - \kappa\delta H)(H d_i(k) - \tilde{\omega}_i).$$

Since $\kappa\delta\|H\|_2 < 1$, then we have $I_2 - \kappa\delta H \succeq 0$ and that $\|I_2 - \kappa\delta H\|_2 \leq \frac{\max(1,\beta)}{1+\beta} \overset{\Delta}{=} c < 1$. Denote $\tilde{err}_i(k) := \|H d_i(k) - \tilde{\omega}_i\|_2$, we have

$$\tilde{err}_i(k) \leq c^k \tilde{err}_i(k).$$

Applying the same technique in proving Eq. 27, we have that $\tilde{err}_i(K) < \frac{e}{2}$ for any $\kappa > \kappa^{(4)}$ for some $\kappa^{(4)} > 0$, which is sufficient to obtain Eq. 28. $\qquad\square$

# G  RECONSTRUCTED NEURAL SIGNALS IN SEC. ??

We visualize the reconstructed signals in Fig. 6. Specifically, if we denote the SWT transformation as $g$, then for the Wavelet Shrinkage that is shown in the left-hand image, we visualize the original signal $y$ (marked by blue), the reconstructed sparse signal $g^{-1}(\theta_\lambda)$ (marked by orange) and the noise $y - g^{-1}(\theta_\lambda)$ (marked by yellow); for our SWDI that is shown in the right-hand image, we visualize the original signal $y$ (marked by blue), the reconstructed strong signal $g^{-1}\theta^s$ (marked by orange) and the weak signal $g^{-1}\theta - g^{-1}\theta^s$ (marked by yellow), and the noise $y - g^{-1}(\theta)$ (marked by purple).

As shown, the sparse signal of the Wavelet Shrinkage shows a large difference from the original signal, especially at the peaks and valleys of oscillations. Therefore, it may miss a lot of information that may be mainly accounted for by the weak signal and the bias due to the threshold parameter. In contrast, the reconstructed strong signal by our SWDI can learn more information due to the ability to remove bias. More importantly, we are pleasantly surprised to find that the reconstructed weak signal shares a similar trend to the original signal. In this regard, it can well capture the pattern encoded in the neural signal. This result suggests that the weak signal, which may refer to the non-burst signal that can encode the conduct effect of the electric field, has a non-ignorable affection on the formation of neural signals. Such an effect, together with the additional information learned by the strong signal of SWDI over the sparse signal of the Wavelet Shrinkage, can well explain the more significant medication response achieved by our method than the Wavelet Shrinkage.

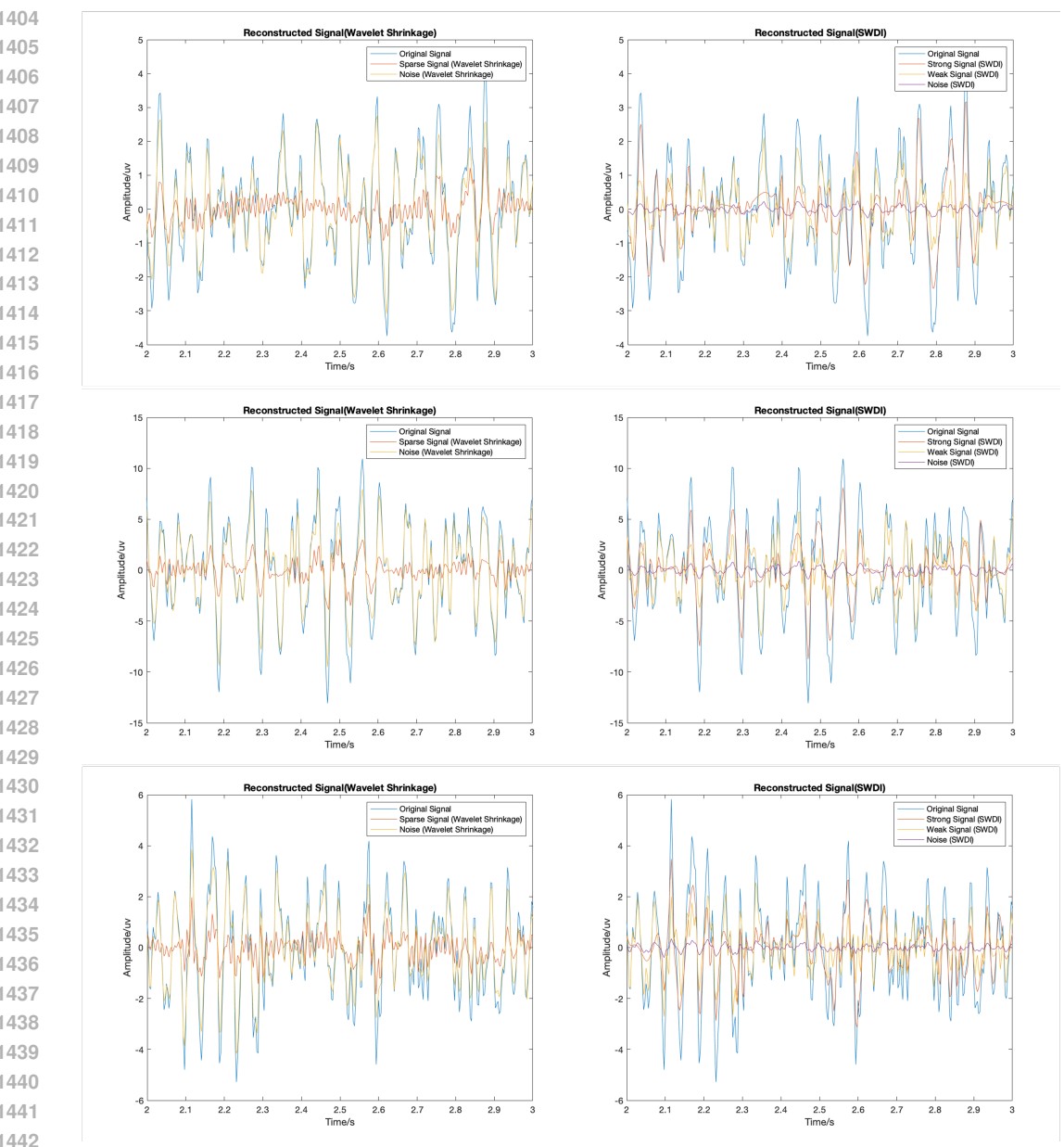

Figure 6: Visualization of reconstructed signals. Left: the original signal $y$ (marked by blue), the reconstructed sparse signal $g^{-1}(\theta_\lambda)$ (marked by orange) by the Wavelet Shrinkage and the noise $y - g^{-1}(\theta_\lambda)$ (marked by yellow); besides the original signal, the right image presents the strong signal $g^{-1}(\theta^s)$ (marked by orange), the weak signal $g^{-1}(\theta) - g^{-1}(\theta^s)$ (marked by yellow) learned by SWDI, and the noise $y - g^{-1}(\theta)$ (marked by purple).

## H  ELECTROENCEPHALOGRAPHY SIGNAL DENOISING

**Data & Problem Description** We extract one subject with 80 trials from Walters-Williams & Li (2011), which comprises a 32-channel Electroencephalography (EEG) signal recorded from a single subject. We added Gaussian white noise to each channel with the signal-to-noise ratio set to 20. We present the mean-squared error (MSE) for the signal reconstruction achieved by our method compared to Wavelet Shrinkage.

**Implementation Details.** Similar to Sec. **??**, we perform a 1-$d$ stationary wavelet transform (SWT) on EEG as the Symlet 8 with level 6. We follow Donoho & Johnstone (1995) to estimate $\sigma$ as

$\tilde{\sigma} = \text{Median}(W_j)/0.6745$. For Wavelet shrinkage, we select $\lambda$ according to the minimax rule in Donoho & Johnstone (1994). For SWDI, we set $\kappa = 20$, $\delta = 1/(\kappa(1+\rho))$ with $\rho = 0.1$ and the stopping time as $\hat{t} = (1 + 1/\rho)/\tilde{\sigma}$.

**Results Analysis.** Fig. 7 shows that our method, especially the dense parameter, can significantly outperform the Wavelet Shrinkage and the sparse parameter, which suggests the importance of capturing weak signals and the capability of our method in learning such weak signals. It's worth mentioning that even when neglecting weak signals, the sparse parameter can still surpass the Wavelet Shrinkage method, thanks to the reduced biases inherent in our methods, as asserted in Thm. D.2 and D.4.

Figure 7: MSE (Microvolts) of the sparse parameter of SWDI $\theta^s(\hat{t})$ (red), dense parameter $\theta(\hat{t})$ (blue) and the Wavelet Shrinkage (black).

