# OpenReview forum: "Splitted Wavelet Differential Inclusion for neural signal processing"
_ICLR.cc/2025/Conference — ICLR 2025 Conference Withdrawn Submission_

### Official Review · Reviewer_owsp · 2024-10-22

**Soundness:** 2
**Presentation:** 3
**Contribution:** 2
**Rating:** 5
**Confidence:** 2

**Summary:**

The paper proposes the Splitted Wavelet Differential Inclusion (SWDI) method for neural signal processing, achieving better strong and weak signal estimation in Parkinson's disease analysis.

**Strengths:**

The paper introduces the Splitted Wavelet Differential Inclusion (SWDI) method, which improves the estimation of both strong and weak neural signals by utilizing an ℓ2 splitting mechanism. It demonstrates better accuracy than traditional wavelet shrinkage, particularly in Parkinson's disease signal analysis, capturing non-burst activity alongside stronger signal components.

**Weaknesses:**

To better demonstrate the proposed method's practicality, I suggest including comparisons with non-wavelet methods, particularly those that have seen recent success in this field. This could provide a clearer perspective on how the method performs in a wider range of real-world applications. Specific examples of non-wavelet techniques, such as deep learning-based methods, would strengthen the evaluation. While wavelet techniques have been widely used in the past, it would be helpful if the authors could justify their choice of wavelets in this context and explain how their approach advances the state-of-the-art. Additionally, comparisons with more recent works in the field would help to clarify the method's relevance and novelty. The paper's content seems more suitable for signal processing journals or conferences, such as TSP, INDIN, or ICASSP.

**Questions:**

It would be beneficial for the authors to discuss the tradeoffs between wavelet techniques and more recent methods, such as deep learning-based approaches, in this specific application. A comparison or discussion of how their method performs relative to recent non-wavelet techniques would provide valuable context for evaluating the method's effectiveness.The distinction between weak signals and noise could be clarified further. I suggest that the authors provide more detailed criteria for how they distinguish between the two, and discuss whether there are alternative approaches. The "differential" aspect of the proposed method requires clearer explanation. A step-by-step description of how the differential inclusion is applied, ideally with a simple example, would greatly improve the clarity of this concept and make it more accessible to readers.

---

### Official Review · Reviewer_BHCx · 2024-10-30

**Soundness:** 2
**Presentation:** 1
**Contribution:** 2
**Rating:** 3
**Confidence:** 3

**Summary:**

This paper considers the longstanding problem of recovering a temporal univariate signal from its noisy observations, which a fundamental problem in signal processing. The authors approach this challenge using wavelet analysis, where they propose to partition the signal into strong and weak components based on the magnitudes of the wavelet coefficients. The paper presents a new method, termed Splitted Wavelet Differential Inclusion (SWDI), which is designed to recover the strong component by employing a differential inclusion framework. It is shown theoretically and empirically in a simulation that the proposed method recovers the strong signal more accurately than other methods based on wavelet shrinkage. Additionally, the method is demonstrated in application to neural signals, where the goal is to identify medication effects on Parkinson’s disease.

**Strengths:**

- The considered problem is central and important.
- The application to neural signals, specifically in the context of Parkinson’s disease, is interesting.

**Weaknesses:**

- The **presentation** of the entire paper, and particularly, the technical aspects is challenging to follow, which hinders comprehension of the core ideas and derivations.
- Due to the presentation style, it is difficult to clearly appreciate the novelty of the paper. The mix of formal and informal statements complicates rigorous validation.
- The **problem setting** lacks clarity, particularly its statistical model. While the strong signal is defined based on the noise standard deviation $\sigma$, the method’s dependence on the signal-to-noise ratio (SNR) or $\sigma$ is unclear. Additionally, it is not specified whether the derivations and results assume Gaussian noise or if the true signal $f$ is deterministic.
- **Numerical results** are limited. Figures are of low resolution with small fonts, making them hard to interpret, especially Figure 1. Expanding the numerical experiments to encompass a broader range of cases and a more comprehensive comparison with alternative shrinkage methods would enhance the paper. The justification for the selected baselines is unclear, considering the prevalence of other methods addressing the same problem.

**Questions:**

Clarification is needed regarding the problem setting, statistical assumptions, and the choice of baseline methods (see weaknesses above).

---

### Official Review · Reviewer_Yxeg · 2024-11-02

**Soundness:** 2
**Presentation:** 2
**Contribution:** 2
**Rating:** 3
**Confidence:** 4

**Summary:**

This paper improves on the well-known wavelet shrinkage approach, introducing a novel method coined "Splitted Wavelet Differential Inclusion (SWDI)". as opposed to wavelet shrinkage, it also takes weak components of the signal to be analyzed in consideration. The effectiveness of SWDI is showed by numerical experiments on data from Parkinson patients.

**Strengths:**

* It is a nice idea to go beyond considering only large wavelet coefficients, i.e., the strong part of the signal.
* The author substantiated her/his novel approach with a theoretical foundation (see, i.e., Theorem 4.6).

**Weaknesses:**

* At present, we have powerful methods based on learning. It is not clear and now even discussed why those are not taken into consideration, There might be good reasons, but this requires a careful discussion.
* The numerical experiments only compare to other model-based methods, mainly wavelet-based approaches. Again, in partiular, learning based methods (DNNs, etc.) need to be used for comparison.

**Questions:**

Please see the weaknesses and reply to those.

**Details Of Ethics Concerns:**

Medical data from Parkinson patients is used.

---

### Official Review · Reviewer_BPQZ · 2024-11-03

**Soundness:** 4
**Presentation:** 2
**Contribution:** 3
**Rating:** 6
**Confidence:** 2

**Summary:**

The paper presents a novel method, the Splitted Wavelet Differential Inclusion (SWDI), for enhancing neural signal analysis, particularly for applications related to Parkinson’s disease. SWDI introduces a dual-parameter approach that estimates both the strong and whole signals simultaneously, addressing limitations of previous wavelet shrinkage techniques. The authors demonstrate that their closed-form solution path improves estimation accuracy for both signal components. This work contributes to the field of neural signal processing by offering a robust framework for analyzing complex neural data, which could have significant clinical applications in neurodegenerative disease research.

**Strengths:**

Originality
The proposed SWDI method creatively combines wavelet analysis with differential inclusion to address limitations in current shrinkage methods by focusing on both strong and weak signals, thus enhancing the detection of signal features important for clinical applications.

Quality
The paper is well-founded with rigorous theoretical analysis that supports the authors' claims.

Clarity
Overall, the paper is clearly written, although some improvements can be made (see 'Weaknesses' section).

**Weaknesses:**

The clarity of presentation of this paper can be improved. There are some English mistakes, subject-verb disagreement, missing conjunction 'and', etc. These should be carefully addressed prior to the publication of this paper.

Examples:

line 052: Add 'and' before 'non-parametric shrinkage'.
line 053: Change 'contains in the signal' to 'contained in the signal'.
line 056: Change 'composed by' to 'composed of'.
line 073: Change 'On the other' to 'On the other hand'.
line 107: Add 'and' before 'non-parametric shrinkage'.
line 114: Change 'have' to 'has'.
line 123: Add 'and' before 'the non-burst component'.
line 373: Change 'includes' to 'include'.
line 377: Change 'the same ... with' to 'the same ... as'.
line 382: Change 'as a contrast' to 'in contrast'.
line 386: Change 'compare to' to 'compared to'.
line 402: Add 'and' before 'then increases'.
line 500: Insert 'be' between 'may' and 'due to'.
line 512: Change 'Fig. 3,. 4' to 'Figs. 3 and 4'.

**Questions:**

How does the SWDI method compare to deep learning-based approaches or other adaptive wavelet techniques in terms of accuracy and computational efficiency?

---

### Note · Authors · 2024-11-19

I have read and agree with the venue's withdrawal policy on behalf of myself and my co-authors.